# Remote Sensing Image-Based Analysis of the Urban Heat Island Effect in Bragança, Portugal

**Cátia Rodrigues de Almeida** [1,2,*] , **Leonardo Furst** [3,4] , **Artur Gonçalves** [3,4] and **Ana Cláudia Teodoro** [1,2]

1   Department of Geosciences, Environment and Land Planning, Faculty of Sciences, University of Porto, 4169-007 Porto, Portugal
2   Earth Sciences Institute (ICT), Pole of the FCUP, University of Porto, 4169-007 Porto, Portugal
3   Centro de Investigação de Montanha (CIMO), Instituto Politécnico de Bragança (IPB), Campus de Santa Apolónia, 5300-253 Bragança, Portugal
4   Laboratório Associado para a Sustentabilidade e Tecnologia em Regiões de Montanha (SusTEC), Instituto Politécnico de Bragança, Campus de Santa Apolónia, 5300-253 Bragança, Portugal
*   Correspondence: up201600831@edu.fc.up.pt

**Abstract:** Urban Heat Islands increase surface temperatures which impact the health and well-being of urban populations. Radiative forcing is impacted by changes to the land surface associated with urbanization that are particularly significant immediately after sunset. This paper aimed to analyze the behavior of UHI in different Local Climate Zones (LCZ) in Bragança city (Portugal), using Air Temperature (Ta), satellite images (Landsat 8), and on-site data. The methodology included a seasonal approach, integrating data with different scales (spatial, radiometric, and spectral) and qualitative and quantitative analyses. Google Earth Engine (GEE) optimized the processing time and computation requirement to generate the Land Surface Temperature (LST) maps. The integration of data with different scales corroborated the complementation of information/analysis and detected the correlation between the Ta and LST. However, the identification of the UHI was compromised due to the time of the passage of Landsat 8, and it was identified as the Urban Cool Island (UCI), a complementary effect of UHI, supporting the results of previous studies and for the use of Remote Sensing (RS) for thermal effects analysis.

**Keywords:** climate change; land surface temperature; landsat; google earth engine; correlation; regression



## 1. Introduction

For human life to be viable in a place, it is necessary to make anthropogenic adaptations, that can suppress the vegetation, soil sealing, albedo variation, and increase the unsealed areas with civil construction [1,2]. These anthropic changes influence the local energy balance and contribute to the formation of the Urban Heat Island (UHI) effect which consists of greater absorption of electromagnetic energy and slower night cooling of urbanized surfaces when compared to surrounding areas with the presence of vegetation [3–5].

Voogt and Oke (2003) [6] suggested a sub-classification analogous to UHI, called Surface Urban Heat Island (SUHI) that, in addition to studying surface temperature differences between urban and rural areas, also addresses temporal variability [7]. This paper draws a parallel between these two related variables that can be used to describe the UHI effect.

The main causes for the formation of the UHI effect are: (i) urban morphology and geometry [8,9]; (ii) thermal properties of the constituent materials and their layouts [10,11]; (iii) the heat storage capacity of building materials (such as solar factor, shading coefficient, irradiance, radiosity, emissivity, absorptance, reflectance, and transmittance) [5,12]; (iv) the change and influence on wind speed, depending on surface roughness; (v) the increased absorption of solar radiation, due to the albedo of some surfaces; (vi) albedo [13]; (vii) Width (W)/Height (H)/Length (L) and urban canyons [14]; (viii) Sky View Factor (SVF) [14];

(ix) urban densification [15]; (x) rugosity; (xi) porosity [16]); (xiv) anthropogenic heat [17]; (xii) energy balance in a building and between buildings [17].

There are several impacts associated with UHI, and the most recurrent are: (i) influence on local microclimate (contributing to an increase in surface temperature, reducing Relative Humidity (RH) and latent heat, and intensifying sensible heat); (ii) changes in the displacement of air masses, precipitation, hydrological behavior (such as displacement of water bodies) [10,18]; (iii) thermal discomfort [19]; (iv) socio-environmental and public health impacts [2,16]; (v) increasing mortalities when combined with natural phenomena (such as Heat Waves (HWs)) [20].

One complementary effect of UHI is the Urban Cool Island (UCI), which may occur as a consequence of multiple factors and is normally felt during the morning [21]. One of the determining factors comes from the fact that the UHI effect is, in some studies, calculated using the difference in temperature between open rural areas, with no obstacles to the penetration of electromagnetic energy, and urban areas where the presence of buildings reduces the contact surface and generate shadows in their surroundings. Under these particular circumstances, the UCI may occur from soon after sunrise until late in the morning [3,22]. The effect has already been identified in previous studies conducted in Bragança, the study area of this paper, in the morning period [23].

Both UHI and UCI effects can be studied through the use of several methodologies, such as Remote Sensing (RS), with the number of publications growing, especially since 2016 [24]. Considering this methodology, satellites equipped with thermal sensors operate in the atmospheric window of the Thermal Infrared (TIR) (8–14 μm, whose absorption rate is minimal), and measure, at the top of the atmosphere, the radiation emitted by the Earth's surface [25,26]. The recorded data varies according to the albedo of the analyzed surface, that is, the absorption or reflection of electromagnetic energy will depend on the constitution of the material, the density of construction, the heterogeneity, etc. [17]. Each sensor has its operational specificities, such as spatial resolution (the pixel size, in meters), spectral resolution (the measure of the width of the spectral bands: the more numerous and narrow the bands, the greater the spectral resolution and the sensor's sensitivity in distinguishing between two levels of intensity of the return signal, which contributes to the production of more accurate spectral signatures), radiometric resolution (the sensor's ability to discriminate small variations in energy, i.e., the higher the radiometric resolution, the more sensitive it is to detect small emitted/reflected differences) and temporal resolution (the period between two consecutive observations of the same point) [27].

In this paper, Landsat 8 data was used. Landsat mission has had a long and uninterrupted observation program, since 1972 [28]. The data are released free of charge by the United States Geological Survey (USGS), at levels 1 and 2 (without pre-processing and with processing, respectively) [29]. OLI sensor (Landsat 8) consists of a panchromatic band and eight multispectral bands, with a resolution of 15 and 30 m, respectively. In the TIRS sensor, there are two thermal bands with a resolution of 100 m, whose data are resampled to 30 m [29]. The band used in this work was band 10, relative to TIRS 1 (10.3–11.19 μm), with a radiometric resolution of 16 bits.

For UHI studies, it is common to use TIR, RGB, and Near Infrared (NIR) bands, the provenience of RS, to understand the Land Use Land Cover (LULC), estimate the LST, and differentiate the Vegetation Index (VI). LST is a variable in the physics of local and global surface processes, linked to radiative, latent, and sensible heat fluxes at the surface. It is known as radiometric or "skin" temperature and consists of the direct measurement of the Earth's surface temperature [30]. The LST will comprise the radiometric temperature aggregate of the set of components, including in the field of view of the sensor [30,31]. The value is recovered by estimating the emitted surface radiance (obtained by an atmospheric correction in the radiance sensor), and by inverse application of the Planck function, considering the effects of emissivity variation [30].

In addition to understanding LST, for UHI studies it is necessary to understand the dynamics of the local urban climate. The formation of the topography of mountainous

regions, such as Bragança, is due to the movement of tectonic plates and erosion processes (which depend on other factors, such as climate) [32,33]. The specific climate is one of the parameters used to distinguish from other systems, such as valley and lowland zones, and is associated with three effects: (i) synoptic weather systems or air flows that alter dynamic and thermodynamic processes; (ii) differences generated in regional conditions (cloudiness, precipitation regimes, dynamic and thermal winds, etc.); (iii) terrain morphologies and slopes [32,33].

Altitude and orography can play a vital role in local variabilities, such as warmer environments on slopes that favor solar exposure or on enclosed valleys during the day, while the same locations may be cold and humid during the night and, particularly, in winter [34].

Mountains can be considered complex for UHI studies due to the variability of some environmental parameters (such as temperature, radiation, precipitation, etc.) in a short period, the high daily and annual temperature range, lower mean annual temperatures, extreme winter behavior, interannual climate variability, and snow cover during the year or in some months [33,35], while there is a challenge in separating anthropogenic from natural influences for the formation of this effect.

Considering the methodological application in other locations [36–38], Bragança (Portugal) was chosen as the study area for this work because it has the following characteristics: (i) there are no previous studies using RS data to analyze the LST at the site; (ii) it has the previous classification of Bragança in Local Climate Zones (LCZ) and UHI studies considering data from air temperature measured by the sensor network (Ta$_{SN}$) [23,32,39], that can be used to validate the data obtained in RS; (iii) its mountainous configuration can influence the circulation and speed of winds corroborating, naturally, to the formation of the UHI; (iv) presence of centralized urban densification with surrounding rural areas.

The objectives of this paper are: (i) compare SUHI and UHI data; (ii) analyze data collection on-site, to evaluate their possible correlations; (iii) apply descriptive and quantitative statistical analyses (correlation and regression model), to analyze the behavior of Bragança's UHI in each LCZ, in summer and winter (seasonality).

## 2. Materials and Methods

### 2.1. Study Area

Located in the extreme Northeast of Portugal's mainland (41°48′20″ N, 6°45′42″ W) (Figure 1), Bragança Municipality has 24,078 inhabitants and an area of approximately 1174 km$^2$. It is located in a mountainous region, with a relief considered complex due to the variability of its elevation [23].

The urban area of Bragança is composed of a diverse urban fabric, including different buildings and vegetation densities. The climate is characterized by high annual and monthly thermal amplitude. Precipitation has characteristics of a Mediterranean climate, with frequent rainfall in the winter and relatively dry summers [23].

Climate change issues are relevant to Bragança, which is one of the municipalities that integrate the ClimAdapt.Local project, launched in 2015, which supports projects that aim to develop adaptations to climate change in Portugal [40]. The "*Bragança: Estratégia Municipal de Adaptação às Alterações Climáticas*" document proposes climate change mitigation actions for several areas, such as agriculture and livestock, water resources, forestry, in addition to proposals for awareness and environmental education of the population and policies that prioritize the green economy [40]. Although the municipality has identified addressing climate change as a priority, additional actions may still be considered to moderate the UHI effects, such as the integration of vegetation or the preservation of ventilation paths in more densely urbanized areas.

In a previous study of the UHI, Bragança was classified into seven different LCZs, following the methodological procedures indicated by Oke [14]: (i) Compact midrise (CM): areas with the existence of modern construction of medium-high height, high density, and paved surfaces—sensors 3, 7, and 13; (ii) Compact low-rise (CLR): the ancient center of

the city, with medium-low height, high-density, built-in rock, and brick—sensors 4 and 6; (iii) Open midrise (OM): medium density, streets of low height dwellings in bands or isolated—sensor 10, 12, 18 and 22; (iv) 8–Large low-rise (LLR): commercial and industrial, low average density with low and high buildings with paved parking—sensors 5, 17, and 21; (v) Urban Green Spaces (GAB): urban green spaces, predominantly green cover with undergrowth and trees—sensors 2, 8, 9, and 11; (vi) Sparsely built (SB): transition space between urban and rural environments, scattered houses with agricultural and forestry surroundings—sensors 1, 14, and 15; (vii) Rural Areas (RCD): isolated rural areas in the suburbs of the city are representative of the characteristics of the local landscape—sensors 16, 19, 20, and 23 [32].

Ta$_{SN}$ is obtained by 23 sensors (model TGP-4500, TinyTag, Gemini Data Loggers, Chichester, UK), was installed in December 2011 (three meters above ground level, encapsulated by white boxes, facing south, and semi-leveled from street lights [23]) and distributed according to LCZs classification, in the urban and peri-urban context (Figure 1). There are at least three sensors in each LCZ, in a dispersed way, considering the urban and rural gradient [16]. For each point, the following parameters were identified: (i) the land use, occupation, and land cover; (ii) the SVF (recorded through photographs—fisheye); (iii) the roughness length [41]; (iv) the fraction of impervious surface, within a radius of 25 m, estimated in ArcGIS software [23].

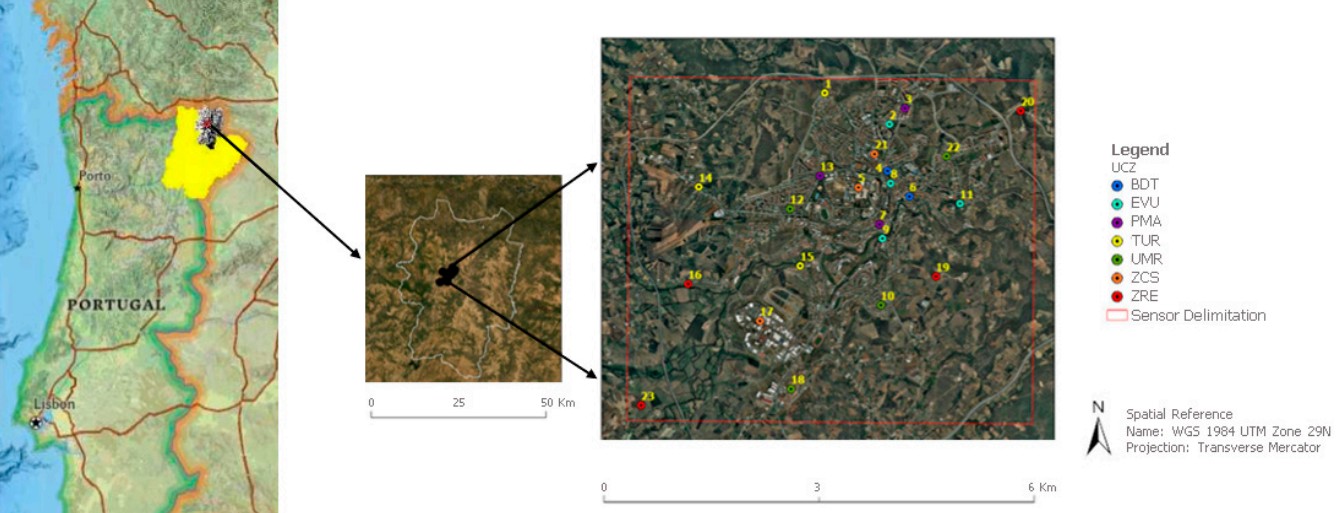

**Figure 1.** Study area, with the sensor delimitation (in red) and the sensor network installation site (23 sensors).

### 2.2. Data Collection

Four on-site data collections were performed: three in summer and one in spring, coinciding with the satellite overpass. It was not possible to perform any collection on-site in the winter due to the weather conditions (precipitation and/or clouds on data) and due to the pandemic period (COVID-19 [42]), since traveling between different districts of Portugal was limited. This limitation does not apply to the LST obtained from Landsat 8 data (LST$_{L8}$) and Ta$_{SN}$ data.

Ta of Bragança were obtained from the website of the Instituto Português do Mar e da Atmosfera [43] and meteoblue [44]. The information is detailed in Table 1.

The data collected used can be divided into: (i) Ta$_{SN}$, collected between 2013 and 2021; (ii) LST obtained with the handheld thermal camera (LST$_{HTC}$) and photographs obtained at 10 points; (iii) Wind Speed—Va (m/s), and Average Radiation Temperature—Tr (°C) measured by four portable monitoring equipment; (iv) LST$_{L8}$ between 2013 and 2021.

**Table 1.** Collection dates and Ta of Bragança.

| Collection | Date (mm/dd/yyyy) | Season | Ta (°C) | | |
|:---:|:---:|:---:|:---:|:---:|:---:|
| | | | Min. | Max. | Mean |
| C1 | 08/03/2020 | Summer | 12.0 | 26.0 | 19.0 |
| C2 | 09/04/2020 | Summer | 12.0 | 34.0 | 23.0 |
| C3 | 04/16/2021 | Spring | 2.0 | 16.0 | 9.0 |
| C4 | 07/05/2021 | Summer | 14.0 | 25.0 | 19.5 |

The definition of the site for the $LST_{HTC}$ measurement followed these criteria: (i) ten locations with diverse spatial distribution were chosen, to contemplate, at least, one sensor of each of the seven LCZs; (ii) the data from the sensor present in the chosen area would serve as validation; (iii) for each of the ten locations, different surfaces were sampled, considering, as much as possible, the local heterogeneity. As a result, 27 different surfaces were sampled, and the respective coordinates were registered.

HT Instruments THT33 were used, with a resolution of $80 \times 80$ pixels, Field Of View (FOV) of $21° \times 21°$, a spectral field between 8–14 μm, and an Instantaneous Field Of View (IFOV) (@1m) of 4.53 mrad. The collection followed a standard height of 1.5 m between the camera and the target (the error associated with the operator was the same for all sampled data). The samples were collected twice at each point: in the morning, between 10 h and 12 h (UTC) (considering the limit of one hour before and one hour after the satellite overpass), and in the afternoon, between 13 h to 15 h (UTC). Photographs were recorded with a camera to assist in the classification of the surfaces, according to their composition.

The portable monitoring equipment (Delta Ohm 32.1 and 32.3) included the sensors: AP3203—Omnidirectional Hot Wire Probe; TP3275–Globe Temperature Probe; HP3217R—Combined Temperature and RH Probe. They were positioned at different points on the days of on-site collection, considering the heterogeneity of the surfaces and the proximity to at least one of the 23 sensors: (i) sensor 17–Industrial Zone (1–LLR); (ii) sensor 9–inside the IPB campus (2–GAB); (iii) sensor 6–"Praça Camões", in a historic square in the city center (3–CLR); (iv) sensor 3–densely urbanized neighborhood (4–CM). Each station was programmed to record data every two minutes, between 10 h and 15 h (UTC). To minimize the effects of shadow in the data collection process, the equipment was placed in sun-exposed locations.

For the $LST_{L8}$ collection, Landsat 8 overpass time in Bragança is at 11 h 15 (UTC), with a temporal resolution of 16 days. To compare the RS data with those obtained by the sensor network, 87 winter and summer scenes were processed between 29 June 2013 and 5 July 2021.

*2.3. Data Processing*

For $Ta_{SN,}$ 87 dates coinciding with the Landsat 8 flyover (in summer and winter) were considered. The mean was calculated between 10 h and 12 h (UTC) (considering one hour before and one hour after the satellite overpass). For the four on-site collections, one additional average was performed, between 13 h and 15 h (UTC).

For $LST_{HTC,}$ adapting methodologies that were used to analyze the UHI on different surfaces [12,45,46], the 27 surfaces were classified into six classes, namely: (i) Asphalt (As); (ii) Sidewalk (S); (iii) Vegetation (V); (iv) Vegetation with exposed Soil (Vs); (v) Dry Vegetation (DV), and (vi) Dry Vegetation and Rock (DVSt) (Table 2). The sampling areas were selected by considering at least one representative from each LCZ (which acts as an urban-level classification).

For the result, the average temperatures (°C) generated by the HTMercury33 application [47] were included.

**Table 2.** Classification of the points sampled with the handheld thermal camera and the equivalent fixed sensor.

| Point | Sensor | LCZs | No. of Surfaces Sampled | Description/Surface Classification |
|-------|--------|------|-------------------------|-------------------------------------|
| P1 | 23 | RCD | 2 | P1A–DV (collection 1, 2, and 4) P1A–V (collection 3) |
|    |    |     |   | P1B–Vs (collection 1, 2 and 4) P1B–V (collection 3) |
| P2 | 18 | OM | 3 | P2A–As (all collections) |
|    |    |    |   | P2B–S (all collections) |
|    |    |    |   | P2C–Vs (collection 1, 2 and 4) P2C–V (collection 3) |
| P3 | 17 | LLR | 4 | P3A–As (all collections) |
|    |    |     |   | P3B–Vs (collection 1 and 2) P3B–V (collection 3 e 4) |
|    |    |     |   | P3C–S (all collections) |
|    |    |     |   | P3D–V (collection 1, 2 and 4) P3D–Vs (collection 3) |
| P4 | 15 | SB | 3 | P4A–AS (all collections) |
|    |    |    |   | P4B–S (all collections) |
|    |    |    |   | P4C–DVSt (collection 1, 2, and 4) P4C–VSt (collection 3) |
| P5 | 9 | GAB | 2 | P5A–Vs (collection 1, 2 and 4) P5A–V (collection 3) |
|    |    |     |   | P5B–V (collection 1, 2 and 3) P5B–DV (collection 4) |
| P6 | 5 | LLR | 5 | P6A–V (collection 1, 2 and 4) P6A–DV (collection 3) |
|    |    |     |   | P6B–As (all collections) |
|    |    |     |   | P6C–S (all collections) [1] |
|    |    |     |   | P6D–S (all collections) |
|    |    |     |   | P6E–S (all collections) |
| P7 | 6 | CLR | 2 | P7A–S (all collections) |
|    |    |     |   | P7B–S (all collections) |
| P8 | 3 | CM | 2 | P8A–S (all collections) |
|    |    |    |   | P8B–As (all collections) |
| P9 | 14 | SB | 2 | P9A–V (collection 1, 2 and 4) P9A–Vs (collection 3) |
|    |    |    |   | P9B–As (all collections) |
| P10 | 22 | OM | 2 | P10A–As (all collections) |
|     |    |    |   | P10B–S (all collections) |

[1] Due to structural changes in the surface, the data from this point were disregarded in the analysis.

$LST_{L8}$

To obtain the $LST_{L8}$, its necessary to convert the Digital Number (DN) into LST. For this task, the Google Earth Engine (GEE) was used, which performs the data processing in a cloud [48], using the collection LANDSAT/LC08/C02/T1_L2. It is already available with the calculated LST, including atmospheric, emissivity correction, and cloud mask application (when necessary) [49–51]. To evaluate if this collection presented reliable values,

the calculation of the LST $_{L8}$ of 4 September 2020 was performed, using the raw Landsat 8 data, following the standardized procedures [31] in ArcGIS software. The result obtained in ArcGIS and GEE were compared and showed an average difference of 0.36 °C, with the largest difference being 1.5 °C in the highest temperatures, which can be associated with the extra corrections applied in GEE [49].

To extract the $LST_{L8}$, the ArcGIS software was used, applying a 30 m buffer around each of the 23 sensors on-site. This criterion was defined to guarantee representativeness, considering the values measured in the radius corresponding to a pixel around the sensor [52].

### 2.4. Statistics Analysis

From the plotting of $LST_{L8}$ maps in GEE, a general analysis of the influence of seasonality on surface temperatures and the discrepancies (if existent) was performed. The $LST_{L8}$ maps generated in this step were bounded to the sensor area and assigned a similar color scale to enable comparison and visual analysis of the products.

To calculate the $SUHI_{Int}$ and $UHI_{Int}$, the mean of the $LST_{L8}$ and $Ta_{SN}$ values were calculated for sensors 16, 19, 20, and 23, which integrate the RCD (rural areas) LCZ, separately. The resulting values were used as a reference to calculate the difference between the registered value in the others LCZs (Equations (1) and (2), respectively). This methodology has already been adopted in other UHI studies [53–55]

$$SUHI_{Int} = LST_{L8(LCZ)} - \overline{LST_{L8(RCD)}} \tag{1}$$

where $SUHI_{Int}$ is the intensity value of the $LST_{L8}$ at each point; $LST_{L8(LCz)}$ is the $LST_{L8}$ value extracted from the buffer at each sensor; $\overline{LST_{L8(RCD)}}$ refers to the mean of the $LST_{L8}$ values from RCD sensors.

$$UHI_{Int} = Ta_{SN(LCZ)} - \overline{Ta_{SN(RCD)}} \tag{2}$$

where $UHI_{Int}$ is the intensity value of the $Ta_{SN}$ at each point; $Ta_{SN(LCZ)}$ is the $Ta_{SN}$ in each sensor; $\overline{Ta_{SN(RCD)}}$ is the average $Ta_{SN}$ registered in RCD sensors.

The results were analyzed with a boxplot in RStudio [56], a methodology commonly applied in UHI studies [57,58].

For the correlation analysis, the Shapiro test [59] was applied to understand the distribution of the $Ta_{SN}$ and $LST_{L8}$. When the results point to a normal distribution, Pearson's correlation method [60] was applied and, for non-normal distribution, Spearman was used [61]. Both tests are commonly used in UHI studies [62,63].

### 2.5. Validation

From the local data collected by the portable thermal camera and the four portable monitoring equipment, an analysis of the thermal behavior of the measured surfaces and a comparison with the values recorded by the Landsat 8 sensor were proposed, safeguarding the specificities and different scales of each technique.

To evaluate if the UHI values measured on each surface (relative to the ten points) were higher in the afternoon period compared to the morning one, according to previous UHI studies [64], the $LST_{HTC}$ was analyzed. Wind speed and Tr were recorded at four points in different LCZs (LLR, GAB, CLR, and CM), to assess the influence of the urban context on air circulation and local radiant temperature. These four points were chosen for their characteristics and relevance to UHI studies (vegetated areas, highly urbanized areas, and industrial areas). The points RCD, OM, and SB were not considered in this stage due to the amount of available equipment.

#### Linear Regressions

Linear regressions, a methodology adopted in other UHI studies [65], were applied to analyze the relationship between $Ta_{SN}$ data (response variable) from the $LST_{L8}$ (explanatory

variable), considering all the sensors and the influence of seasonality, in RStudio [56,66]. In this paper, the $r^2$ adjusted was used in these analyses.

## 3. Results and Discussion

### 3.1. $LST_{L8}$ Mapping

As expected, seasonality influences the behavior of the city's surface temperature, as the temperatures observed in summer are significantly higher than in winter, as has been identified in studies conducted in some cities in China [38,67,68]. Figure 2 shows an example of a map of each season–31 July 2013 (summer) (A) and 1 March 2016 (winter) (B) and a map from 8 February 2014 (winter) (C), whose result showed values below −5 °C in the analyzed area. All generated $LST_{L8}$ maps are shown in Appendix A (Figure A1).

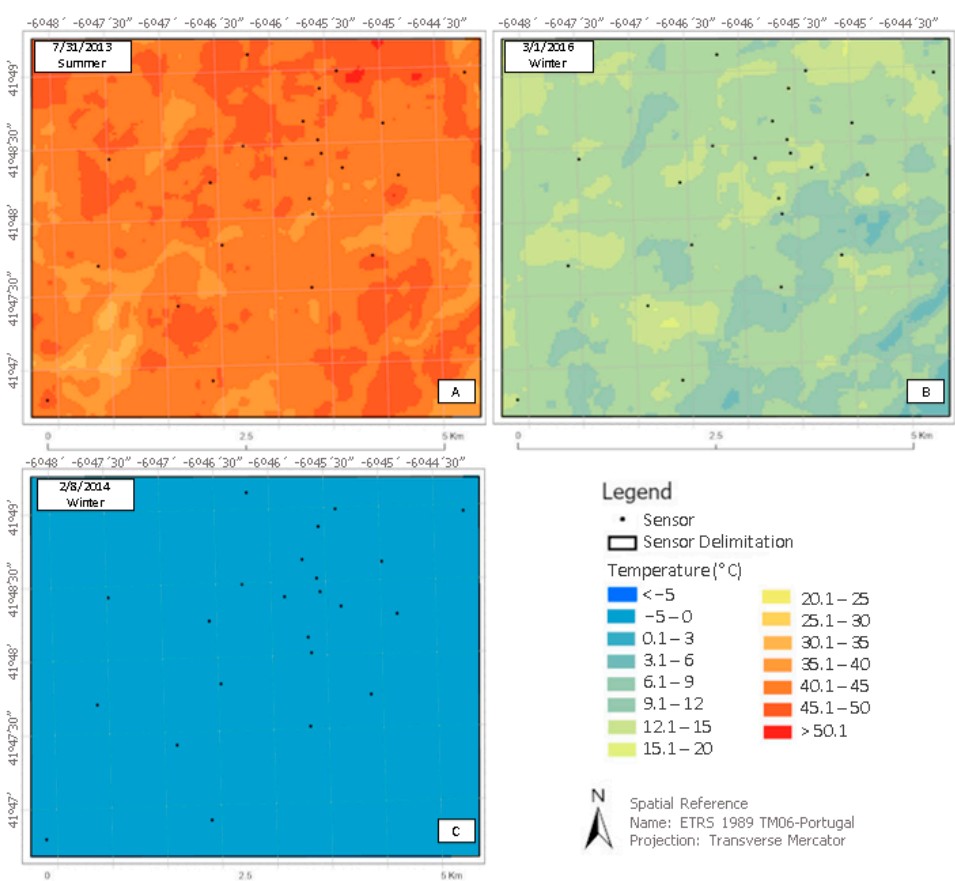

**Figure 2.** Example $LST_{L8}$ maps, calculated/generated in GEE, with the area delimitation of the 23 fixed sensors (black square) and the respective sensors plotted (black dots). (**A**): relative to 31 July 2013 (summer); (**B**) relative to 1 March 2016 (winter); (**C**) relative to 8 February 2014 (winter), which exemplifies the results with values below −5 °C in the study area.

When results fall below −5 °C in at least one of the 23 sensors (Figure 2C), the data for that day were disregarded. Thus, of the 87 dates/scenes processed, 17 were ignored. Of the 69 images processed, 43 were from summer and 26 from winter. This negative value may be associated with the presence of dense clouds that influence data registration, even with the application of a mask and standard methodologies [69]. Of the remaining 70 dates, 69 were considered in all statistical analyses. The image referring to 16 April 2021, was considered only for the field data validation stage because it was the only processed image of the spring.

$SUHI_{Int}$ and $UHI_{Int}$

From the boxplot analysis considering the time of the satellite's passage, both the $SUHI_{Int}$ (Figure 3) and $UHI_{Int}$ (Figure 4) showed similar behavior for the local LCZs. The data from the $UHI_{Int}$ in summer showed fewer outliers than those of the $SUHI_{Int}$, and in winter, the $UHI_{Int}$ showed more amplitude in the data, especially for the GAB locations, compared to the $SUHI_{Int}$.

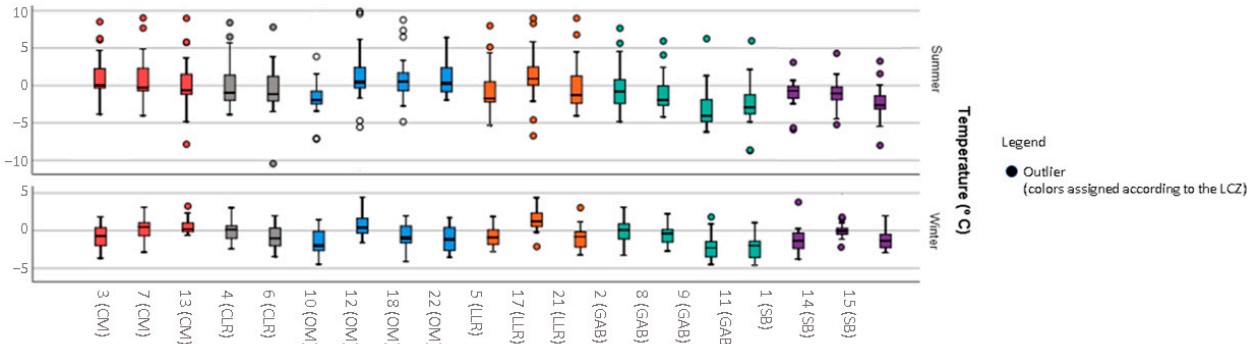

**Figure 3.** Boxplot with the $SUHI_{int}$, calculated for the measurements taking place in the summer and winter, at each sensor and its respective LCZ (the circles refer to the outliers, colored according to their LCZ).

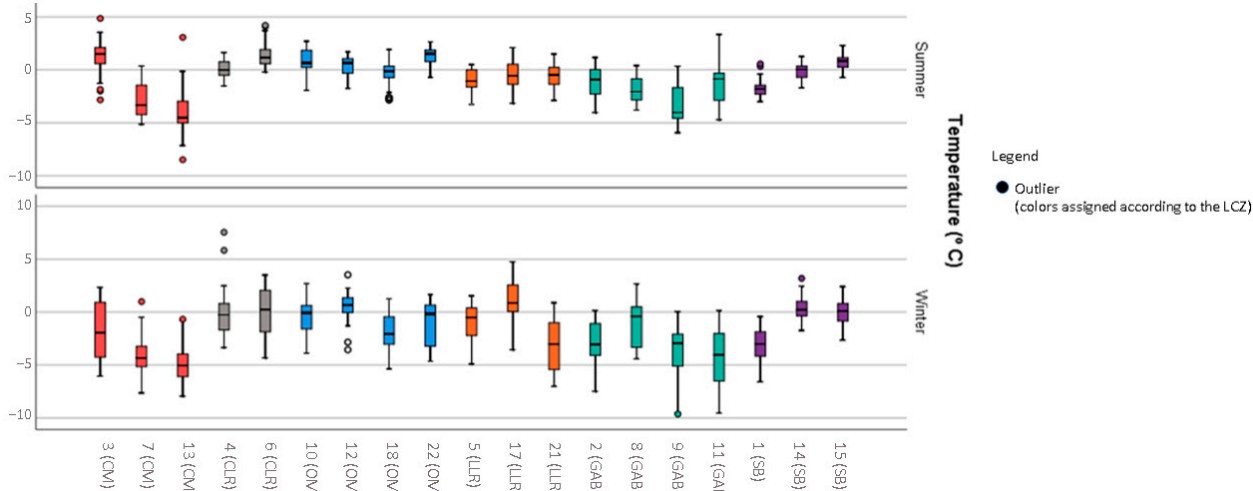

**Figure 4.** Boxplot with the $UHI_{int}$, calculated for the measurement taking place in the summer and winter, at each sensor and its respective LCZ (the circles refer to the outliers, colored according to their LCZ).

CM locations (in red) showed medians near zero or negative for $SUHI_{Int}$ and $UHI_{Int}$. CLR locations (in gray) presented median values of $SUHI_{Int}$ close to zero, in $UHI_{Int}$ (winter and summer). In the $SUHI_{Int}$, the behavior was similar, despite presenting median values slightly below 0 °C (near −2.00 °C in sensors 4 and 6 in summer and sensor 6 in winter). Both locations have anthropically altered elements and similar behavior for the UCI effect.

OM locations (in blue) presented median values of $UHI_{Int}$ also close to 0.00 °C, except for sensor 22 in the summer, with a median close to 2.00 °C, and sensor 18, in winter, which presented a median close to −3.00 °C. As for the $SUHI_{Int}$, the summer median was around 0.00 °C in all sensors, except for sensor 10, which was around −2.00 °C. The median in winter was close to −2.00 °C, except for sensor 12, whose value was close to 0.00 °C. The anthropic elements in these locations are less dense, which is reflected in higher temperatures than observed in the CM and CLR.

LLR locations (in orange) the median $UHI_{Int}$ was close to $-2.00\ °C$ in the summer. In the winter, the behaviors were distinct: near $-1.00\ °C$ at sensor 5, $1.00\ °C$ at sensor 17, and $-3.00\ °C$ at sensor 21. As for the $SUHI_{Int}$, in the summer, sensors 5 and 21 showed median values near $-3.00\ °C$ and sensor 17 around $1.00\ °C$. In winter, sensor 17 maintained its behavior, and sensors 5 and 21 showed similar results at $-1.00\ °C$. Since this is an area with commercial/industrial buildings, the temperature differential is higher, similar to CM and CLR.

GAB locations (in green) showed negative $SUHI_{Int}$ and $UHI_{Int}$ values in both winter and summer, with the highest air temperature amplitudes recorded in winter and the lowest median in sensor 9.

SB locations (in purple) the $UHI_{Int}$ was varied in winter and summer, but recorded results were close to $0.00\ °C$ (exceptions in sensor 1, which presented $-3.00\ °C$ in winter and summer, and in sensor 15, which presented $2.00\ °C$ in summer). As for the $SUHI_{Int}$, the data presented more outliers, especially in the summer. The median results in both stations ranged between $0.00$ and $-2.00\ °C$, except for sensor 15, which showed a median of $-3.00\ °C$ in the summer. The behavior of the SB sites was close to the RCD, considering that the boxplot is more homogeneous compared to the other CZUs, which may be associated with the presence of vegetation at the site (considering that it is a transition area between urban and rural spaces).

As for the maximum $UHI_{Int}$ in the summer, it was recorded by the CLR (sensor 6) followed by CM (sensor 3). In the winter, the highest $SUHI_{Int}$ were recorded at the LLR (sensor 17) and CLR (sensor 6). The highest values of the $SUHI_{Int}$ in the summer were recorded by the OM locations (sensors 12 and 22). In the winter were recorded by OM (sensor 12) and LLR (sensor 17).

The lowest $UHI_{Int}$ values in the summer were recorded by the CM (sensor 13) and GAB (sensor 9). In the winter, the lowest $SUHI_{Int}$ were recorded by locations GAB (sensors 9 and 11) and CLR (sensor 6). In the $SUHI_{Int}$, the lowest values in the summer were recorded by GAB (sensor 9), SB (sensor 15), and LLR (sensor 5). In the winter, by GAB (sensors 9 and 11) and OM (sensor 10).

In a general analysis, lower $SUHI_{Int}$ and $UHI_{int}$ values were observed in more anthropically altered areas. As the Landsat 8 overpass occurs near the solar nadir (11 h 15 UTC), which minimizes the projection of shadows, it was expected that the UCI effect would have been less evident. However, considering that the rural vegetated areas had more exposure to electromagnetic energy in the early morning hours compared to urban areas, these conditions have determined the presence of higher $LST_{L8}$ and $Ta_{SN}$, thus determining a negative $UHI_{Int}$. These results are similar to preliminary studies that identified UCI in Bragança [23], and RS data came to corroborate the same results.

In a study carried out in Iraq, UCI was also identified using RS data obtained by Landsat 8 in the morning period: open spaces, airports, and low-density housing in the peripheral regions of the city showed higher LST values [70], just as occurred in the peripheral and less dense rural areas in Bragança.

### 3.2. Correlation

The Shapiro test results for $Ta_{SN}$ and $LST_{L8}$, considering all data, i.e., without separation by season, indicated a non-normalized behavior in all sensors (*p*-value $< 0.05$), which justifies the application of the Spearman correlation method. Considering all dates (without separation by season), data shows a non-normal distribution, therefore Spearman's statistical method was adopted.

The same test was applied considering the summer and winter seasons. In the summer, the same non-normalized behavior was observed for $LST_{L8}$ data. The behavior of the $Ta_{SN}$ data followed a normal distribution. As the assumption of the Pearson method is that all data that will be used in the processing present normality, the Spearman method was also applied in the summer.

In the winter, the data mostly presented *p*-values > 0.05, for $LST_{L8}$ except for three sensors. To make the analysis coherent and to apply the appropriate method in data analysis, Pearson's correlation was adopted (even for the data that did not present a *p*-value > 0.05 $LST_{L8}$ in sensors 9, 16, and 19), to minimize possible discrepancies that could be associated with the difference of the analytical method in the same sample group. The normalized behavior of the winter data may be associated with a smaller temperature range, due to the specific environmental conditions of winter, which affect the net solar radiation available [71], and the smaller number of observations, compared to the summer data. In summary, Spearman was the method applied for all data and summer data, and Pearson was applied to winter data.

### 3.3. Correlation between $Ta_{SN}$ and $LST_{L8}$

Although $LST_{L8}$ and $Ta_{SN}$ are different parameters and that weak correlations can be expected due to the different external factors affecting them [65], all correlations were positive, and, considering all days, the results were classified as "Strong" and "Very Strong". In summer, 17 sensors (73.91%) had a "Strong" correlation, and six sensors (26.09%) had a "Very Strong" correlation. In winter, all correlations were "Very Strong", which may be associated with lower solar incidence and shadow projections compared to summer (Figure 5).

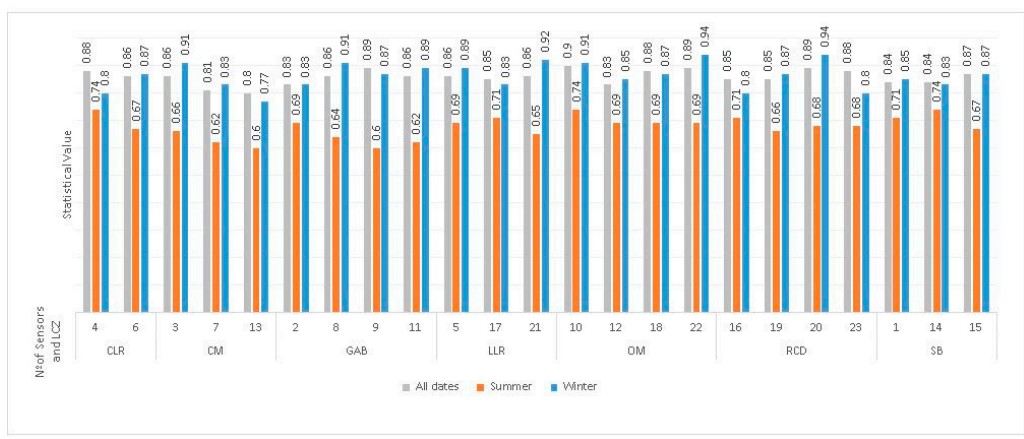

**Figure 5.** Statistical values were obtained from the correlation between $Ta_{SN}$ and $LST_{L8}$ (absolute values).

In summer, the incidence of solar energy is more intense and the parameters of $Ta_{SN}$ and $LST_{L8}$ are affected differently. The Landsat 8 pass time is close to the solar nadir, but there may still be shadows casting on the surface, resulting in lower correlations in this season compared to winter. These shadows may be associated with: (i) the configuration of buildings, which in periods of greater availability of electromagnetic energy can influence the result obtained by comparing a shaded area to another exposed, unobstructed area; (ii) the vegetation, especially trees as a consequence of the canopy cover, who will provide shadow. Unlike most urban structures, some trees (deciduous) will present changes in their capacity to intercept solar radiation, due to the changes in the foliage around the year, with notable influence on the microclimate in its surroundings [9,71].

The local sensors record the $Ta_{SN}$ in a microscale context, while the $LST_{L8}$ is obtained covering a more extended scale, closer to the level of generalization used for the definition of the LCZs [72]. RS was able to identify patterns in the thermal behavior of the different LCZs in Bragança, which are correlated to the $Ta_{SN}$.

The correlation between Ta and LST has already been identified in other studies and locations, such as in South Korea and Brazil [73,74], which corroborates the methodology and the relationship between both indicators for UHI analysis.

### 3.4. On-Site Collection Data for Validation

The classification of the surfaces measured with the handheld thermal camera was performed and, in Table 2 there is the identification of the area, the proximity fixed sensor,

the classification of LCZs, the number of surfaces collected in each point, and the respective description, based on the data obtained by the photo camera. Due to seasonality, some points with the presence of vegetation had their surface classification changed throughout the measurements taken.

### 3.4.1. Analysis of the $LST_{HTC}$ and $LST_{L8}$

$LST_{HTC}$ allows detailed thermal data from different spatial resolutions, which enables a more specific evaluation of the spectral responses of the surface materials [75]. Data collection is performed point by point, an hourly difference that can be influenced by weather changes, such as the presence/intensification of clouds or shadows, which can influence the net radiation reaching the analyzed surface [9,76].

The revisit time also presents specificity between the techniques: while Landsat 8 performs acquisitions every 16 days at the same local time (UTC) and under different environmental conditions, the on-site collections are more flexible and can be performed at shorter intervals (depending on the availability of financial, human and equipment resources). However, the satellite revisits time and also ensures a data history, that can be important to UHI studies [28].

$LST_{HTC}$ sampled on each surface in the morning compared to $LST_{L8}$ showed higher temperatures in most cases. The justification for this behavior may be associated with the special resolution of the collection, since the LST averages the thermal values existing in a pixel, while the handheld thermal camera presents a smaller FOV, with the ability to present more details of the sampled surface [24].

Furthermore, there are different influencing factors between both techniques, such as the atmosphere that affects satellite data, i.e., the presence of clouds, while casting possible shadows, is less impactful in local collections.

Overall, vegetated surfaces showed lower temperatures than anthropic ones, and, in addition to LULC, the configuration of the surroundings exerts influences on the results, since densely urbanized areas can corroborate with air circulation obstacles [9,77].

P6 (LLR) (Figure 6), with the presence of vegetation and anthropic areas, will exemplify the thermal behavior. There were five different surfaces, and two correspond to Sidewalks (S) (both are in gray—P6D and P6E) (Figure 7). The vegetation was classified in V/DV due to the seasonality (details in Table 2).

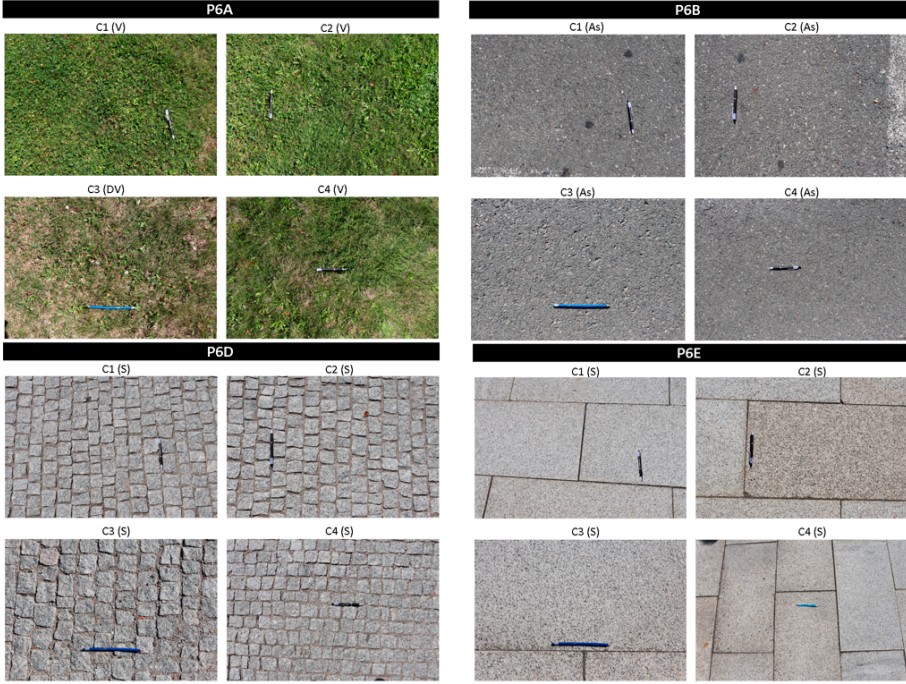

**Figure 6.** Surface sampled in P6 (LLR) and their respective classifications.

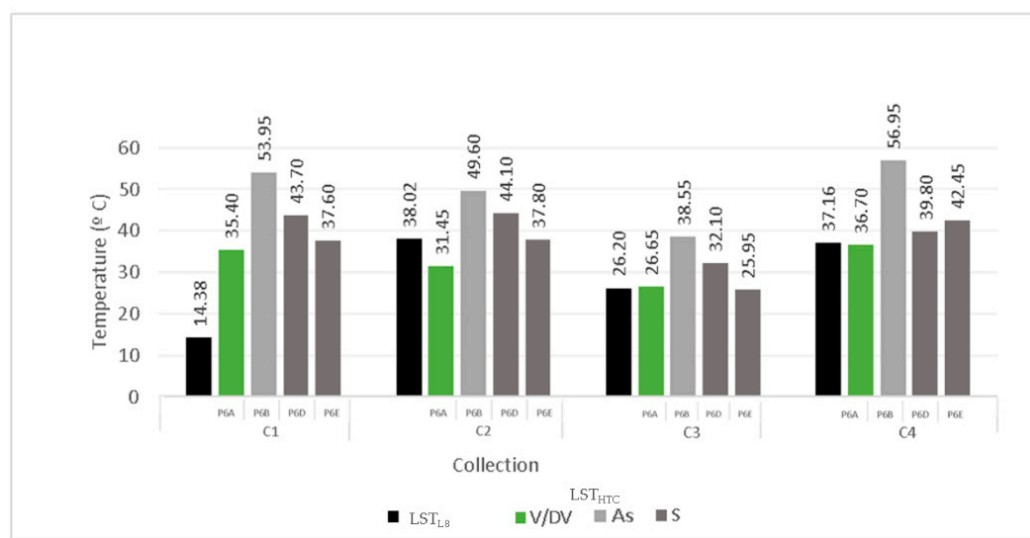

**Figure 7.** Analysis of the temperatures on the various sampled surfaces at P6 (LLR).

The values recorded on the vegetation surfaces were the lowest in all collections, compared to the anthropic surface materials (As, S and S), except for C3 in which the area sampled D showed slightly lower values, which can be associated with the seasonality (spring)/ availability of electromagnetic energy [71]. The highest temperatures were measured on asphalt [12,78] (Figure 7).

### 3.4.2. Analysis of $LST_{HTC}$ Differences between Morning and Afternoon Collections

In general, the thermal behavior indicated higher temperatures in anthropically altered surfaces compared to those with vegetation. Moreover, on C2 and C4 with the presence of clouds, the anthropic areas were more affected than the vegetated ones, which corroborates that clouds/shadow effects have a direct influence on thermal behavior.

The results for P2 (OM) (Figure 8) exemplify this behavior: where there is low-density construction, a higher thermal behavior is observed in the afternoon, except for P2C (Vs), in C3 and C4, whose result was similar in both morning and afternoon. The $LST_{HTC}$ in the Vs showed greater variation between collections in C2 (summer) (Figure 9).

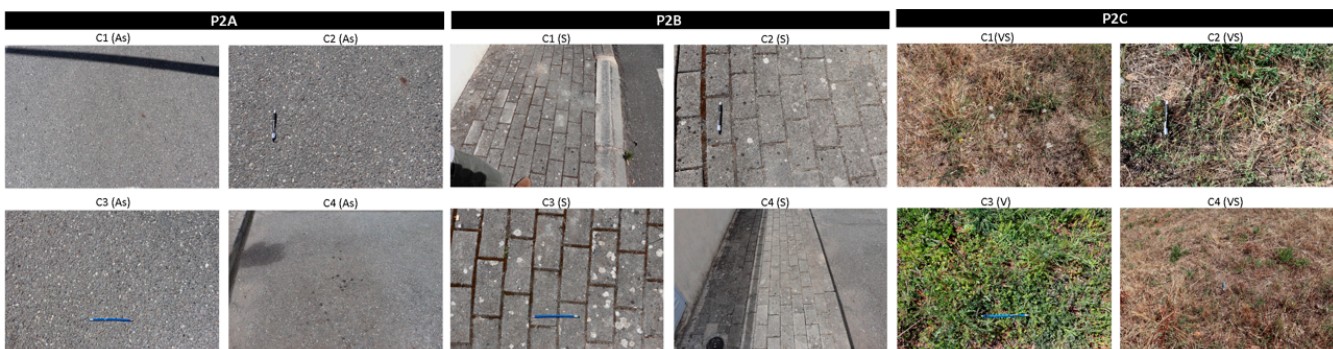

**Figure 8.** Surface sampled in P2 (OM) and their respective classifications.

### 3.4.3. Analysis of $Ta_{SN}$ Differences between Morning and Afternoon Collections

$Ta_{SN}$ showed variability between the on-site collections in most of the points, with a higher range in the afternoon period. This behavior can be justified by the accumulation of electromagnetic energy that occurs throughout the day and that enhances the UHI effect in places with a considerable density of urban construction, especially after sunset [64].

P8 (CM) exemplifies this behavior: the maximum value of $Ta_{SN}$ of 31.15 °C (afternoon), which can be associated with anthropic construction, which may have contributed to the

increase in Ta$_{SN}$ due to surfaces materials (mostly low albedo), and obstacles regarding wind circulation and heat dissipation [79]. A large amplitude of Ta$_{SN}$ is observed, which may be associated with the existence of trees that can act as a natural obstacle to air circulation and, consequently, to the dissipation of absorbed heat (Figure 10) [80,81].

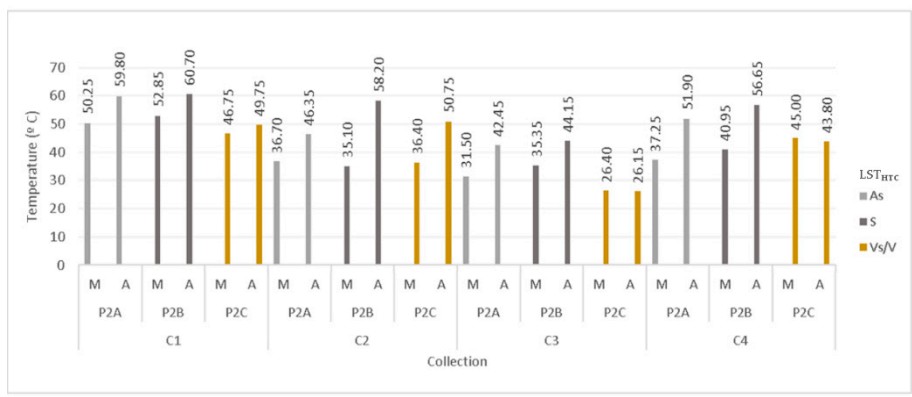

**Figure 9.** LST$_{HTC}$ recorded at Morning (M) and Afternoon (A) collection, at each surface of P2 (OM).

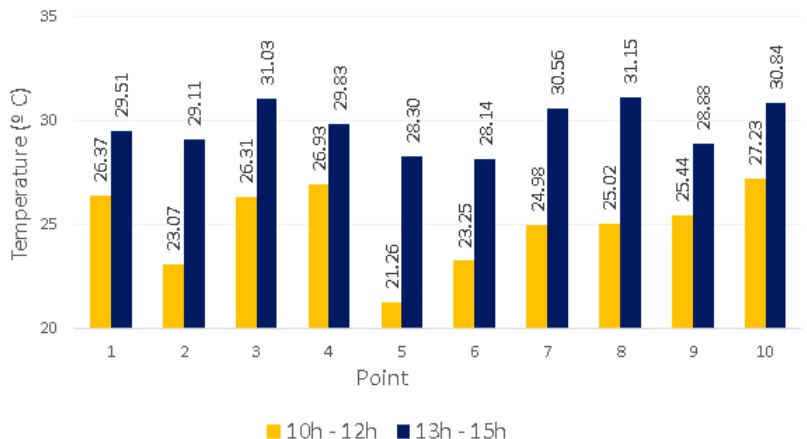

**Figure 10.** Analysis of the Ta$_{SN}$ data collected in the morning and afternoon of collection 1 (3 August 2020—Summer).

### 3.4.4. Wind Speed Analysis (Portable Monitoring Equipment)

C4 (5 July 2021) showed the highest values for wind speed in all portable monitoring equipment, with the highest results observed at station 4 (CM) (Figure 11). This behavior may be associated with the Venturi effect, recorded in street canyon contexts [82] (Figure 12).

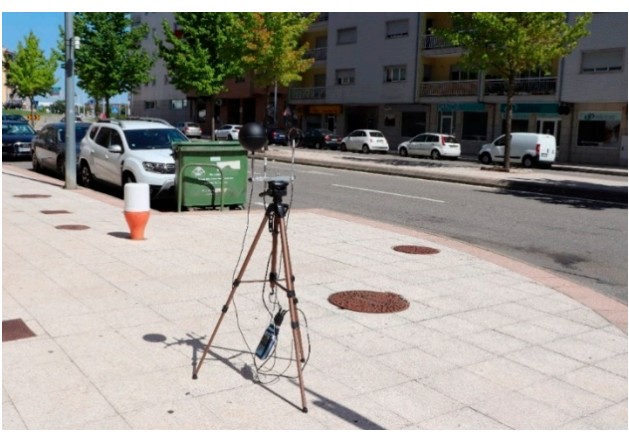

**Figure 11.** P4—Portable monitoring equipment and the surroundings involved.

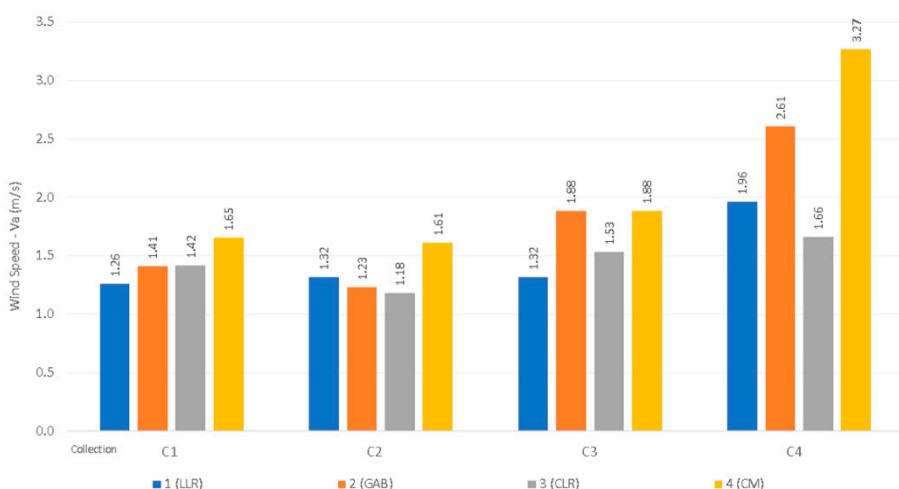

**Figure 12.** Comparison of wind speed at each portable monitoring equipment, per collection day.

The radiation emitted by walls is long-wave while the incoming (solar) radiation is short-wave and is also influenced by the orientation and reflective power of the surfaces. Therefore, the SVF influences the energy exchange at a site and is associated with the W and H of the canyon [83]. When considering a street, the temperature may be somewhat higher than in a more open location, due to the trapping of solar radiation, which is intensified during the night period. If the ambient flow is approximately perpendicular to the axis of a street canyon, a vortex can form, which will carry the ambient air down and the canyon air up the canyon. This mechanism causes the wind speed to be enhanced and, in the case of places with air contamination, can serve to transport the polluting elements to other places [83,84]. For this characteristic to be confirmed at this point, it is recommended that complementary field collections be carried out.

3.4.5. Tr Analysis (Portable Monitoring Equipment)

There is an equivalence between the values measured in each of the collections, with the highest temperatures from the summer measurements, confirming the influence of seasonality on a local scale, because Tr is a consequence of the thermal radiation released from surrounding elements [85].

In the C4, the presence of clouds possibly affected the results, especially CM (P4), which showed the highest values in C1, C2, and C3. Being an area with medium and high-density construction, this behavior may be associated with the retention of solar energy in this LCZ (Figure 13).

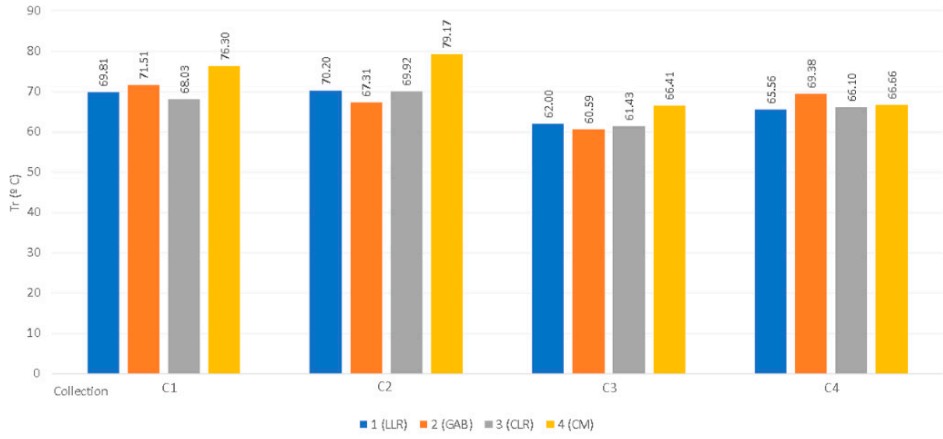

**Figure 13.** Comparison of Tr at each portable monitoring equipment, per collection day.

The portable monitoring equipment 1 installed in the LLR, showed similar results in all collections, except for C3, which may be associated with seasonality. As for the GAB, it is observed that the recorded data were similar to the data from LLR (P1) and CLR (P3), except in C4, which presents the highest temperature of this collection (69.38 °C). In C4, lower results than those observed in the other summer collections were observed, which may be associated with the presence of clouds on that day. The station installed in the CLR showed temperature regularity in the collections performed (Figure 13).

### 3.5. Linear Regression between $Ta_{SN}$ and $LST_{L8}$

The *p*-value results were lower than 0.05, i.e., therefore $Ta_{SN}$ is a significant parameter for the model. The adjusted coefficient of determination ($r^2$ adjusted) demonstrates that, when using all dates, it is possible to predict $Ta_{SN}$ from the $LST_{L8}$ in all sensors, with explainability starting at 65% in sensor 13 (CM) (Figure 14).

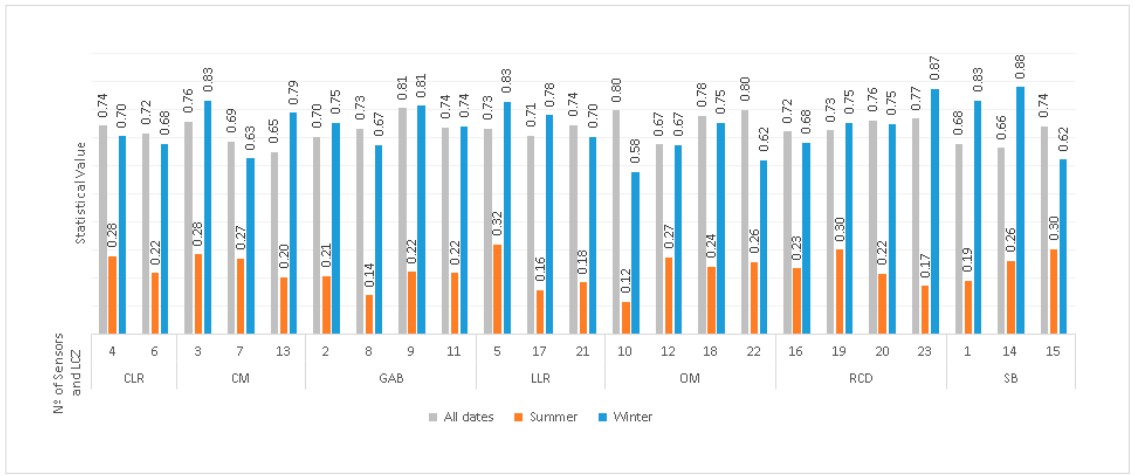

**Figure 14.** $r^2$ adjusted referring to the linear regression model between $Ta_{SN}$ and the $LST_{L8}$.

Considering all dates, CLR sensors presented 75 and 72% (sensors 4 and 6, respectively). SB sensors presented 68, 66, and 74% of explainability results (sensors 1, 14, and 15). Comparing both results, the behavior may be associated with the distinct surface composition in each of the classes: SB has elements of transition between urban and rural areas, and CLR, a former traditional city center, has more heterogeneous anthropogenic surfaces, which may influence the $LST_{L8}$ and $Ta_{SN}$ results more distinctly (Figure 14).

Sensors 2, 8, 9, and 11 (GAB) showed 70, 73, 81, and 74% of explainability for all dates. Sensors 3, 7, and 13 (CM) presented results of 76, 69, and 65%, respectively (all dates). The explainability of GAB is higher than CM and can be associated with the specific surfaces: GAB is composed of green spaces (more homogeneous albedos), compared to CM, which has anthropogenic surfaces, which can affect the $Ta_{SN}$ and $LST_{L8}$ results in specific/distinct ways (Figure 14).

For all dates, sensors 10, 12, 18, and 22 (OM) showed explainability of 80, 67, 78, and 80%, respectively. Sensors 5, 17 and 21 (LLR) resulted in 73, 71 and 74% explainability, and sensors 16, 19, 20 and 23 (RCD) accounted for 72, 73, 76 and 77%, respectively. These three LCZs showed similar behavior in the results, with the highest representatives observed in the OM sensors (except for 12), which have isolated building surfaces of medium density.

Considering the seasonality, the explainability of $Ta_{SN}$ is lower in summer than in winter (Figure 14). In addition, a large part of the sensors in the same LCZ presented variable data. This behavior may be associated with the smaller number of observations and the greater homogeneity of the data in winter, compared to those in summer.

Sensor 10 (OM) was the one that presented the lowest explainability, both in summer and winter (12 and 58%, respectively). Sensors 1, 14, and 15 (SB) showed results of 83, 88, and 62% in winter and 19, 26, and 30% in summer, respectively.

In the CLR sensors (4 and 6), in summer, the $Ta_{SN}$ could be explained from the $LST_{L8}$ with 28 and 22%, respectively. In winter, the results were 70 and 68%, respectively. Regarding sensors 2, 8, 9, and 11 (GAB), the results were 21, 14, 22, and 22% in summer and 75, 67, 81, and 74%, respectively. In the winter results, despite being higher, were less homogeneous than in the summer. Sensors 3, 7, and 13 (CM) showed results of 28, 27, and 20% in summer, and 83, 63, and 79% in winter. Sensors 10, 12, 18, and 22 (OM) showed explainability of 12, 27, 24, and 26% in summer and 58, 67, 75, and 62% in winter. Sensors 5, 17, and 21 (LLR) resulted in 32, 16, and 18% in summer and 83, 78, and 70% in winter.

*3.6. UHI Mitigation Actions*

Mitigation actions for the UHI effect should be supported by studies that identify its causes and which technologies are available/viable from an environmental, social, and economic point of view. The understanding of building structures, LULC, local climate characteristics, and socioeconomic data is relevant for a more assertive public policy proposition [86].

A methodology that can be adopted is the development of a local bioclimatic model that uses urban elements strategically to minimize heat storage, and includes actions such as: (i) inclusion and adjustments of built-up areas to decentralize urban densification; (ii) inclusion and/or maintenance of water bodies, wetlands and/or vegetated areas in urbanized zones, favoring heat exchange between different surfaces; (iii) preservation and/or revitalization of parks, water bodies, and vegetated surfaces (prioritizing densities, tree canopy sizes, and species-appropriate for the location that one wishes to remediate surface temperature); (iv) prioritizing the use of building materials that favor heat exchange rather than heat retention, such as the use of permeable asphalt, for example [24,87,88].

Environmental awareness, urban planning [24], identification of areas with more intensified UHI behavior, change in materials used for construction and/or urban greening, encouraging the use of public transport [89], and tools that can alert residents about heat waves [90] are other examples of mitigating strategies.

To make a specific mitigation proposal for Bragança, aligned with the ClimAdapt.Local project [40], it is necessary to continue this work, overcome some identified limitations and acquire additional information that will allow a deeper analysis of the UHI effect, such as socio-environmental data.

## 4. Conclusions

Studies based on climate change and the analysis of its impacts on microclimates show a growing trend. The UHI, an effect that increases temperatures in places with a higher density of urban construction compared to the surroundings and that can promote impacts on the health and welfare of a population, has been increasingly studied and associated with public policy measures focused on its mitigation.

Since it is an effect with transversal influence, its identification, and analysis, especially in mountainous regions, as in the case of Bragança, becomes a methodological challenge since its formation is associated with several direct and/or indirect factors. For instance, the mountainous configuration of this city, added to the urban density, especially in the city center, corroborates the intensification of the effect, which has been previously identified through scientific research developed in the region, using data from the sensor network.

In this study, data from several sources were used, such as RS, sensors (with continuous data collection), and on-site collections. The integration of different sources and levels of information allows for the analysis at different scales. When using $LST_{L8}$ compared to $LST_{HTC}$, different spatial scales are involved. The RS offers a less detailed spatial scale, due to the pixel resolution, and in transition zones between two different LULC (such as vegetation and urbanization areas), the record considers the average spectral behavior contained in the pixel, which may influence the identification of site-specific and representative spectral contrasts.

Aiming to complement earlier studies carried out in Bragança, this research used a multivariate methodology, including the use of GEE software, RS, and sensor network data from 2013 to 2021, with additional on-site data (portable monitoring equipment and handheld thermal camera), in addition to qualitative and quantitative analysis of the UHI, at different spatial scales. For the collection on-site, the selection of sites prioritized the diversity of LCZs (established in previous studies).

The computation of $LST_{L8}$ in GEE optimizes the computer's processing time and capacity. The collection used with the application of atmospheric and cloud corrections adds value to the process, since it allowed the use of standardized raw data, with minimization of climatic influences and random effects that could be associated with the calculation of each scene, individually.

The integration of different types of data made it possible to obtain more detailed information about the study site and to overcome some limitations inherent to using a single technique, with the addition of complementary results, for example, using the $LST_{HTC}$ data with lower IFOV than $LST_{L8}$, it was possible to obtain a higher spatial detail, especially in the transition zones with different LULC.

Seasonality (winter and summer) was identified in all data. This behavior can be justified as a function of the amount of electromagnetic energy available for exchange with the environment, which reaches greater amplitudes during the summer.

Although the $Ta_{SN}$ and $LST_{L8}$ are different parameters and a strong correlation between them is not necessarily present, since the result of the $LST_{L8}$ depends on the characteristics of the surface, how heterogeneous is the site, and their respective albedos, the results showed "strong" and "very strong" correlation in all sensors. In the summer, approximately 74% of the results showed a "strong correlation" and 26% showed a "very strong" correlation between these parameters. In winter, all data showed a "strong correlation". The observed differences point out the influence of seasonality, where the $Ta_{SN}$ and the $LST_{L8}$ are more convergent when considering winter measurements. Considering that the UHI is a phenomenon affected by several parameters, the availability of electromagnetic energy in the summer for longer periods and at a solar angulation that corroborates energy storage, can be a justification for this behavior. In addition, as the number of observations considered in winter was lower than in summer, this may have influenced the results. Since there are considerable correlations (strong and very strong), RS data/techniques proved to be an effective complementary methodology for UHI studies in Bragança.

As for the UHI, it was not possible to identify this effect due to the time of the Landsat 8 overpass in Bragança, which occurs near the solar nadir (11 h 15 UTC). For the UHI effect to be evident in the results, the collection of thermal data from the night period, provided by satellite missions (thermal sensor or microwave) or by local data collection campaigns, with a UAV with a thermal camera, should be considered.

However, the UCI was identified in the $UHI_{Int}$ results for both $LST_{L8}$ and $Ta_{SN}$, i.e., the effect is observed at different analytical scales, since the $Ta_{SN}$ refers to the micro-scale and $LST_{L8}$ related to a level closer to the LCZ scale. This equivalence ratifies the possibility of integrating RS data in UHI studies, which presented itself as a possible technique for validation, with field data.

The correlation of the $UHI_{int}$ data from $LST_{L8}$ was "very strong" and positive, which may indicate that the differences in thermal behavior are associated with surface albedo, LULC, roughness, permeability, and other urban parameters.

As for the local measurements, although none of them were performed in winter, due to the limitations imposed by the pandemic and the weather conditions, seasonality can be assessed, on a local scale, by considering the differences between the three summer collections and one in spring.

The LST analysis obtained by the thermal camera showed higher temperatures in anthropically altered surfaces when compared to vegetated areas (even dry vegetation). The afternoon collections showed mostly higher temperatures when compared to the morning ones, especially on the anthropically altered surfaces.

It is worth noting that wind speed is a relevant parameter from the point of view of heat dispersion. Tr also followed the same behavior and did not present significant differences in the results, although station 4 (PMA) presented the highest temperatures relative to locations 1 to 3.

Regarding the linear regression and the determination coefficient, it is observed that the air temperature is a relevant parameter in the equation (considering the calculated *p*-value) and that it is possible to obtain its estimated values from the LST data. The summer data present lower predictive/explainability values compared to winter.

Studies of urban climate transcend urban geometry and surface materials and include the landscape interactions, dynamics of housing, commerce, industry, and vehicle circulation, which corroborates the complexity and the need to develop public policies aimed at minimizing the impacts that can be caused to the local population and the environment.

**Author Contributions:** C.R.d.A. participated in the manuscript design, analysis, data processing in software, investigation, validation, and writing—original draft preparation. L.F. participated in the manuscript design, data collection, and processing. A.C.T. participated in the manuscript design, formal analysis, investigation, validation, supervision, and systematic review. A.G. participated in the manuscript design, formal analysis, investigation, validation, supervision, and systematic review. All authors have read and agreed to the published version of the manuscript.

**Funding:** This work was funded by National Funds through the FCT-Foundation for Science and Technology and FEDER, under the projects UIDB/04683/2020 and FCT/MCTES (PIDDAC): UIDB/00690/2020, UIDP/00690/2020 and LA/P/0007/2020.

**Institutional Review Board Statement:** Not applicable.

**Informed Consent Statement:** Not applicable.

**Data Availability Statement:** Not applicable.

**Acknowledgments:** The authors are grateful to the Foundation for Science and Technology (FCT, Portugal) and FEDER for financial support through national funds under the projects UIDB/04683/2020, FCT/MCTES (PIDDAC) to CIMO (UIDB/00690/2020 and UIDP/00690/2020) and SusTEC (LA/P/0007/2020).

**Conflicts of Interest:** The authors declare no conflict of interest.

**Appendix A**

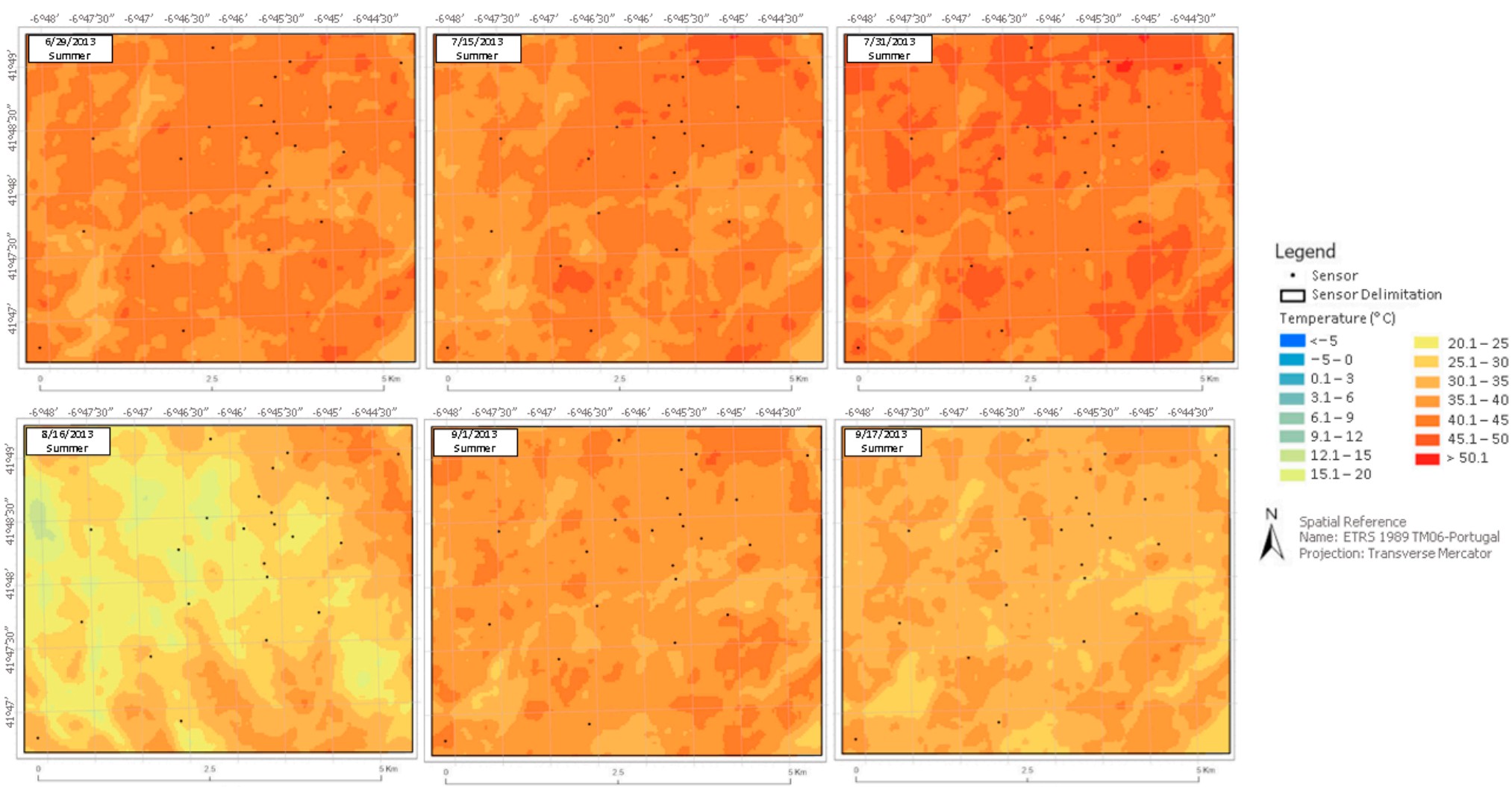

**Figure A1.** *Cont.*

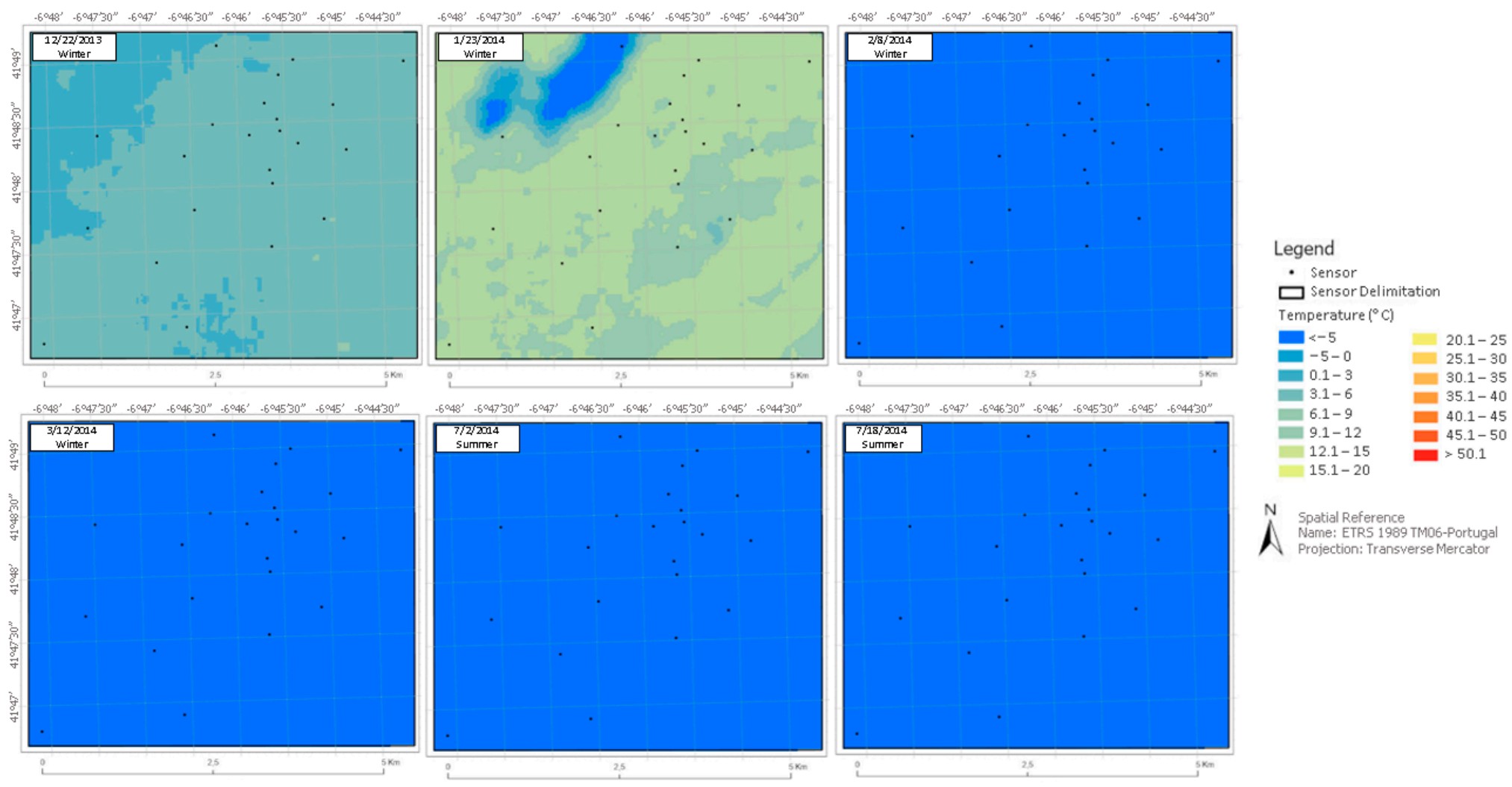

**Figure A1.** *Cont.*

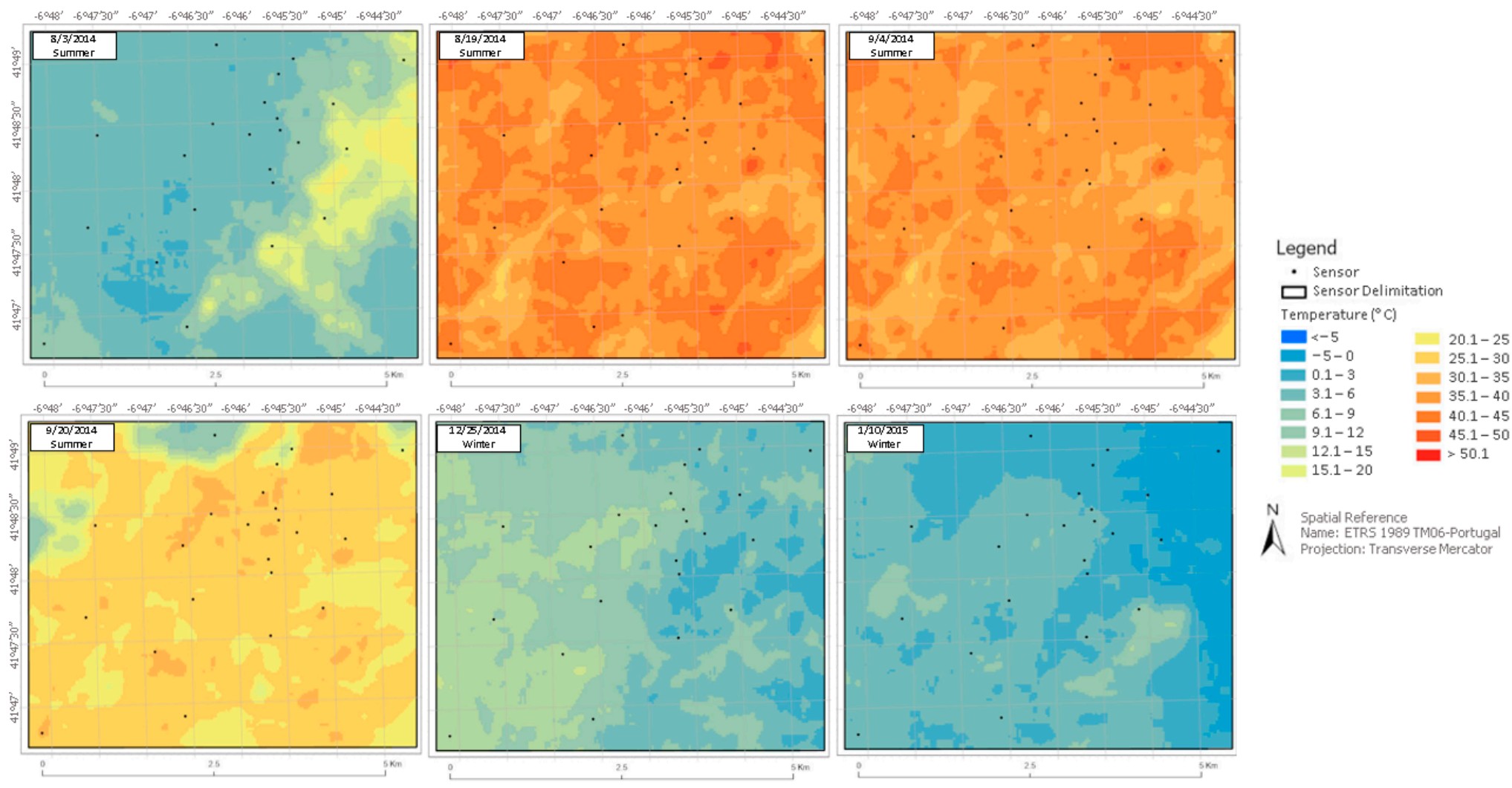

**Figure A1.** *Cont.*

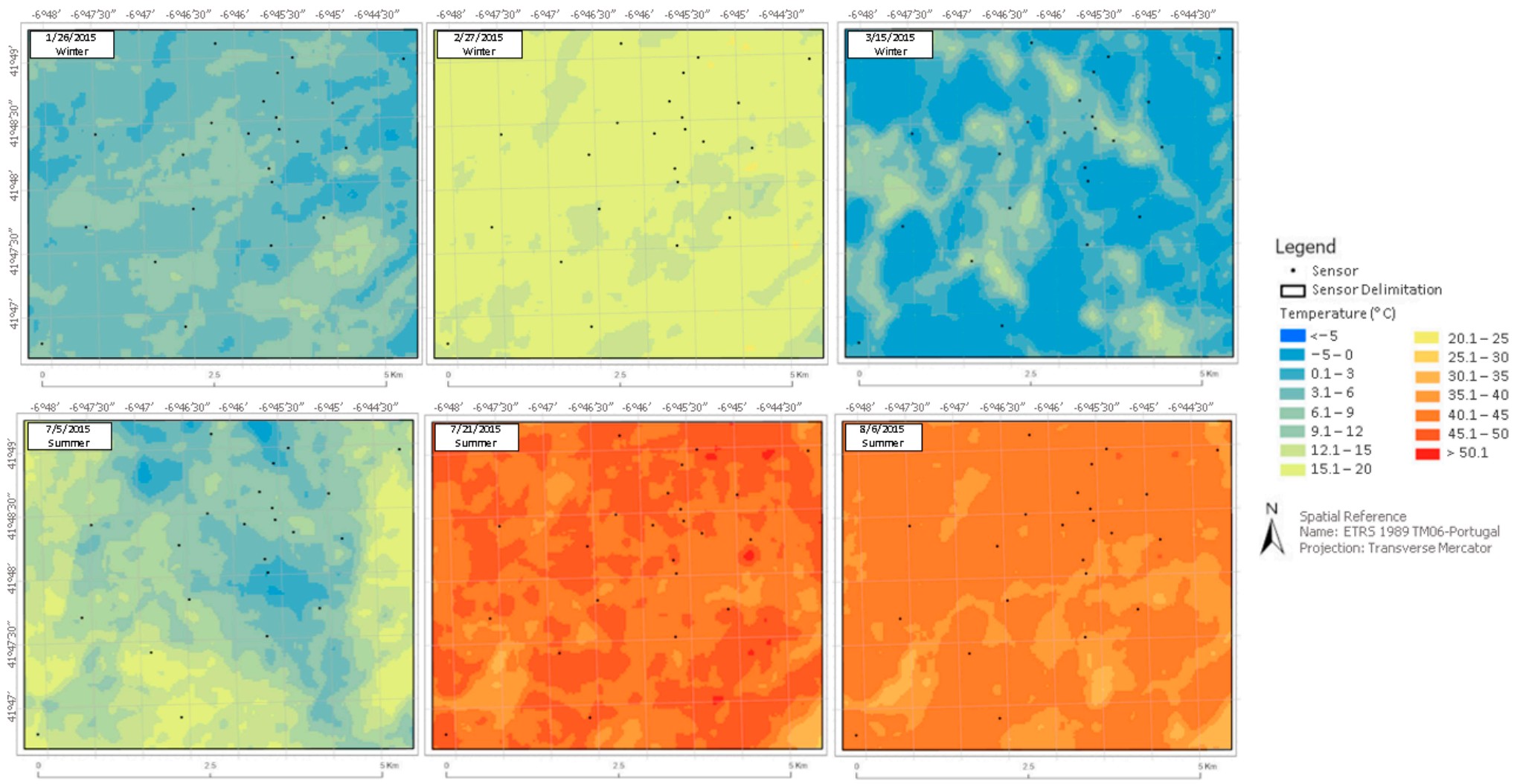

**Figure A1.** *Cont.*

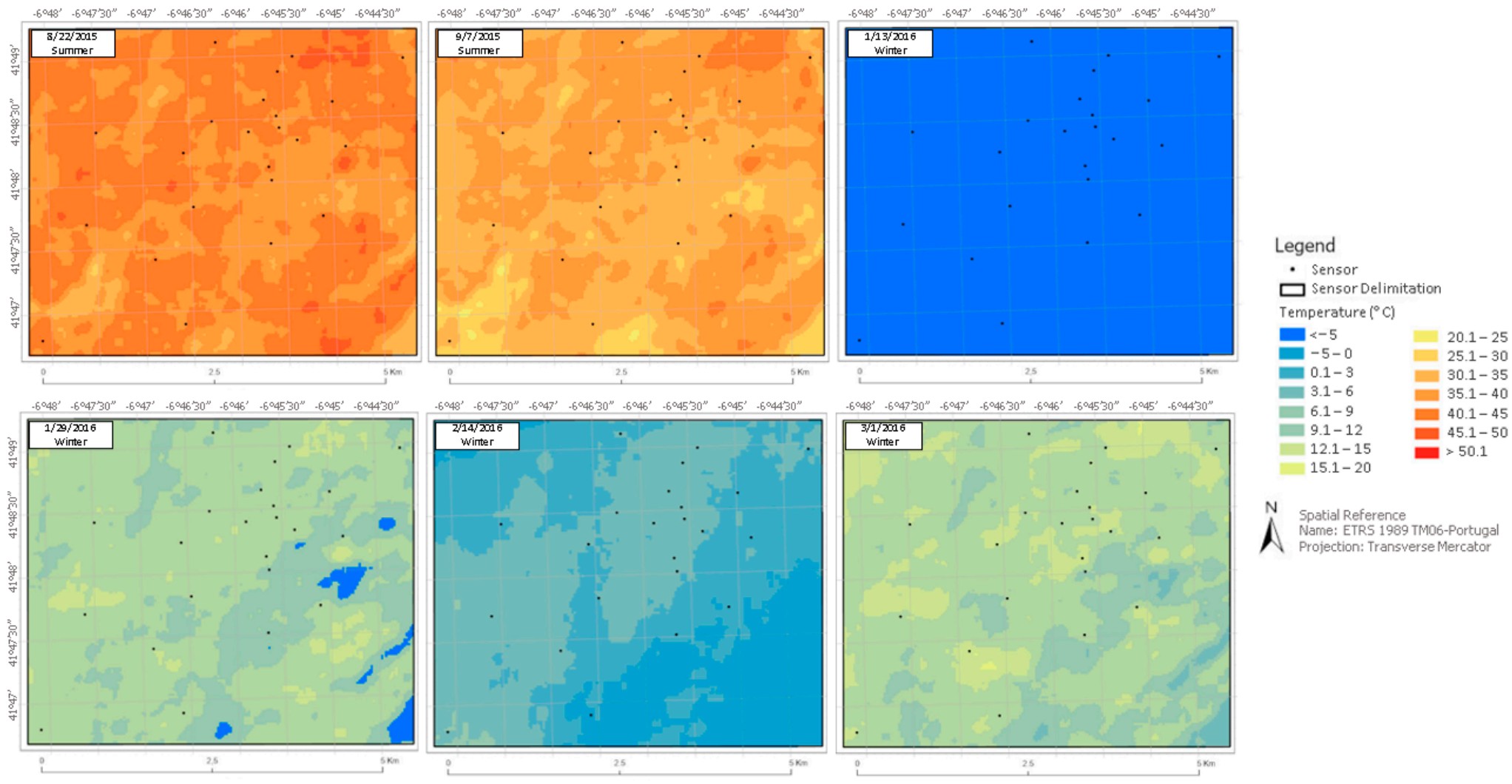

**Figure A1.** *Cont.*

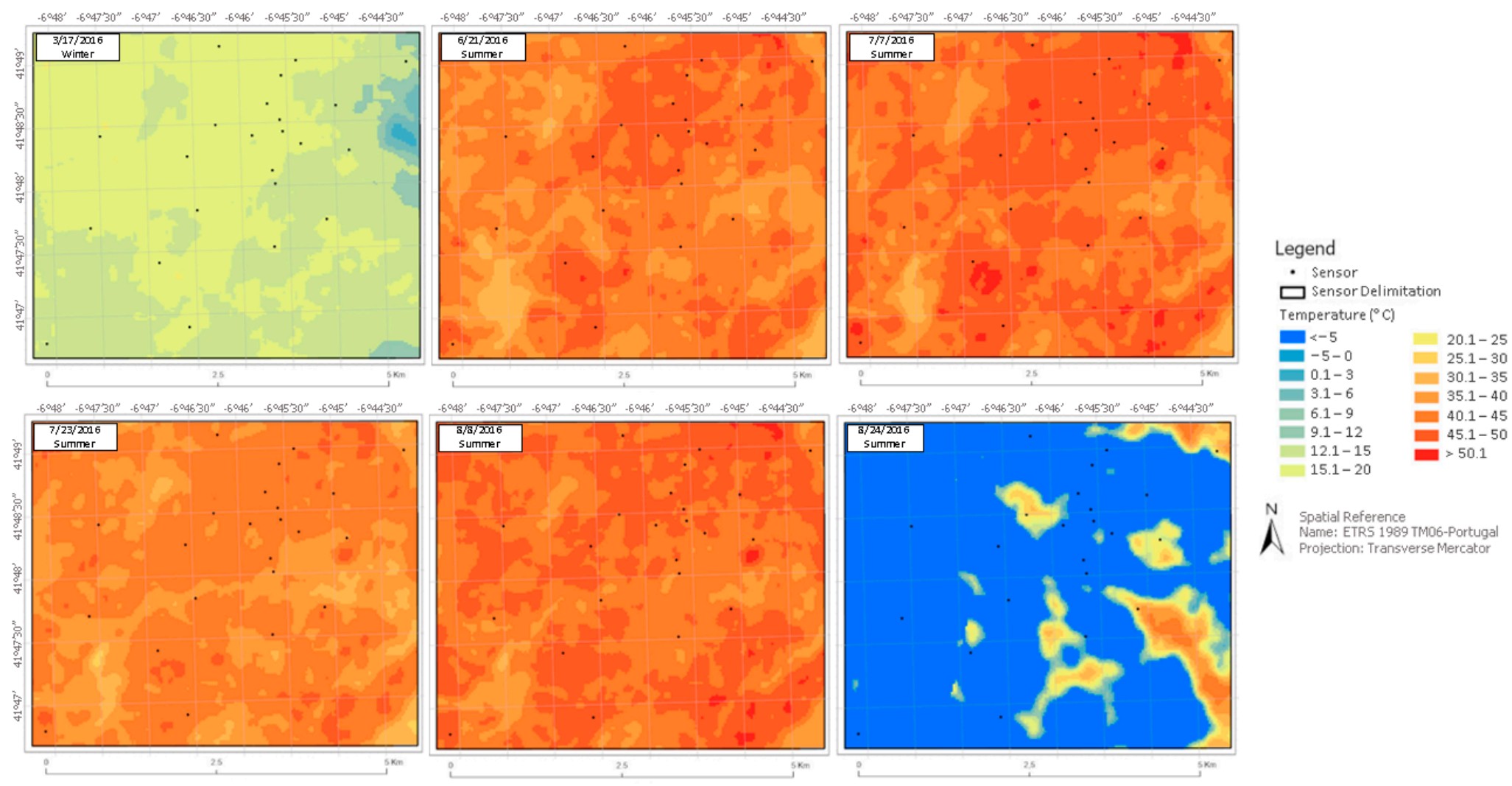

**Figure A1.** *Cont.*

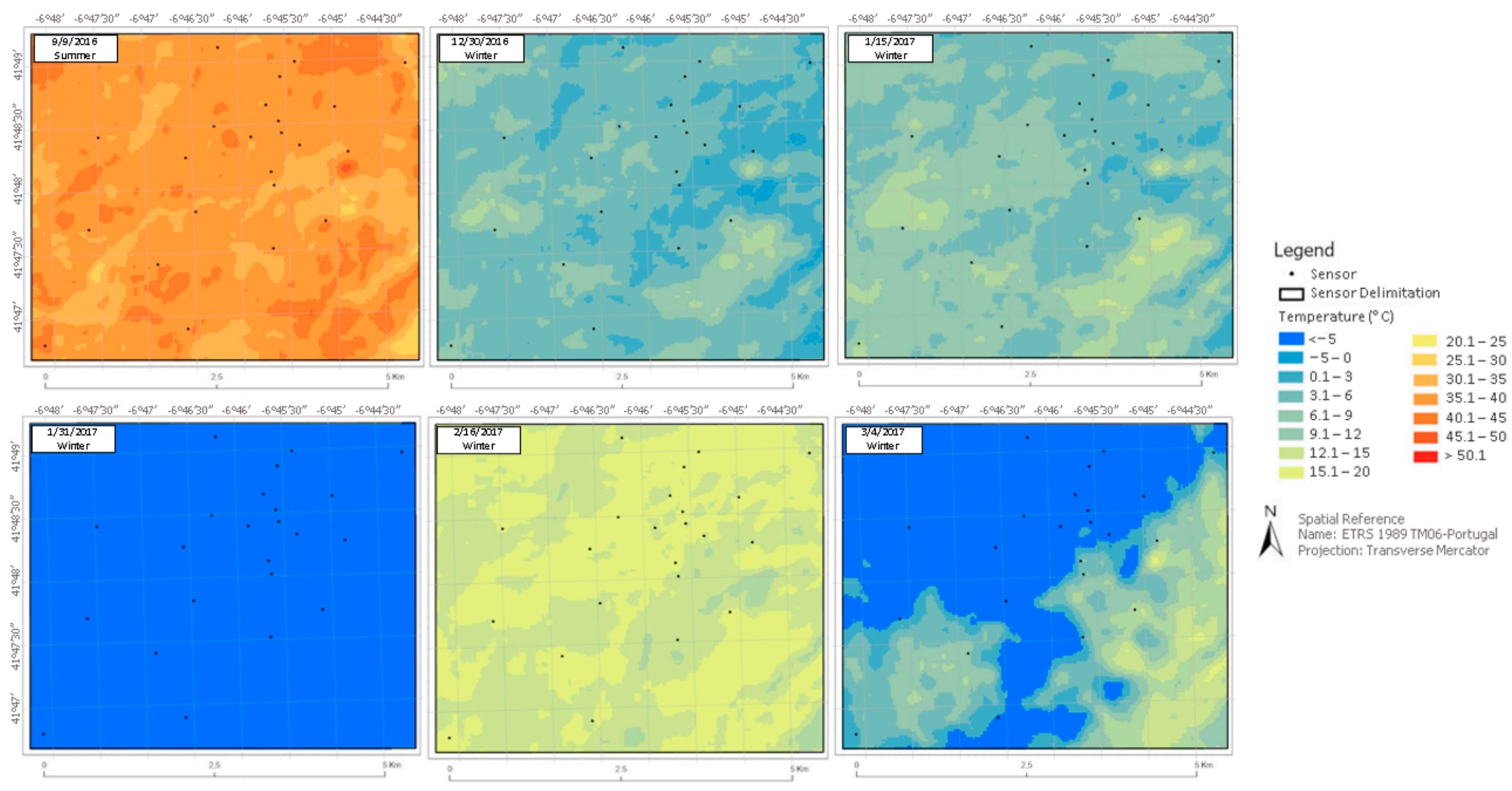

**Figure A1.** *Cont.*

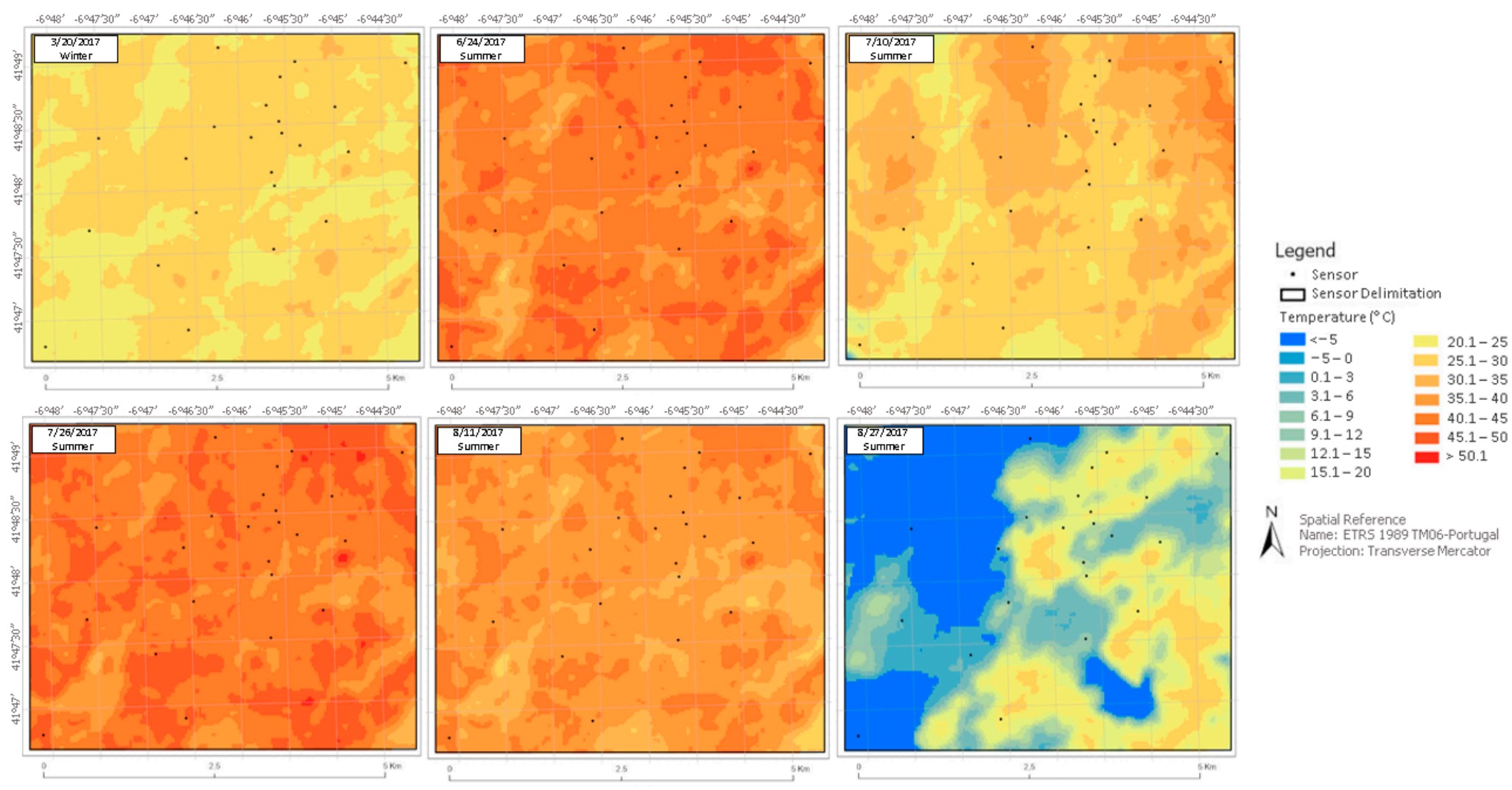

**Figure A1.** *Cont.*

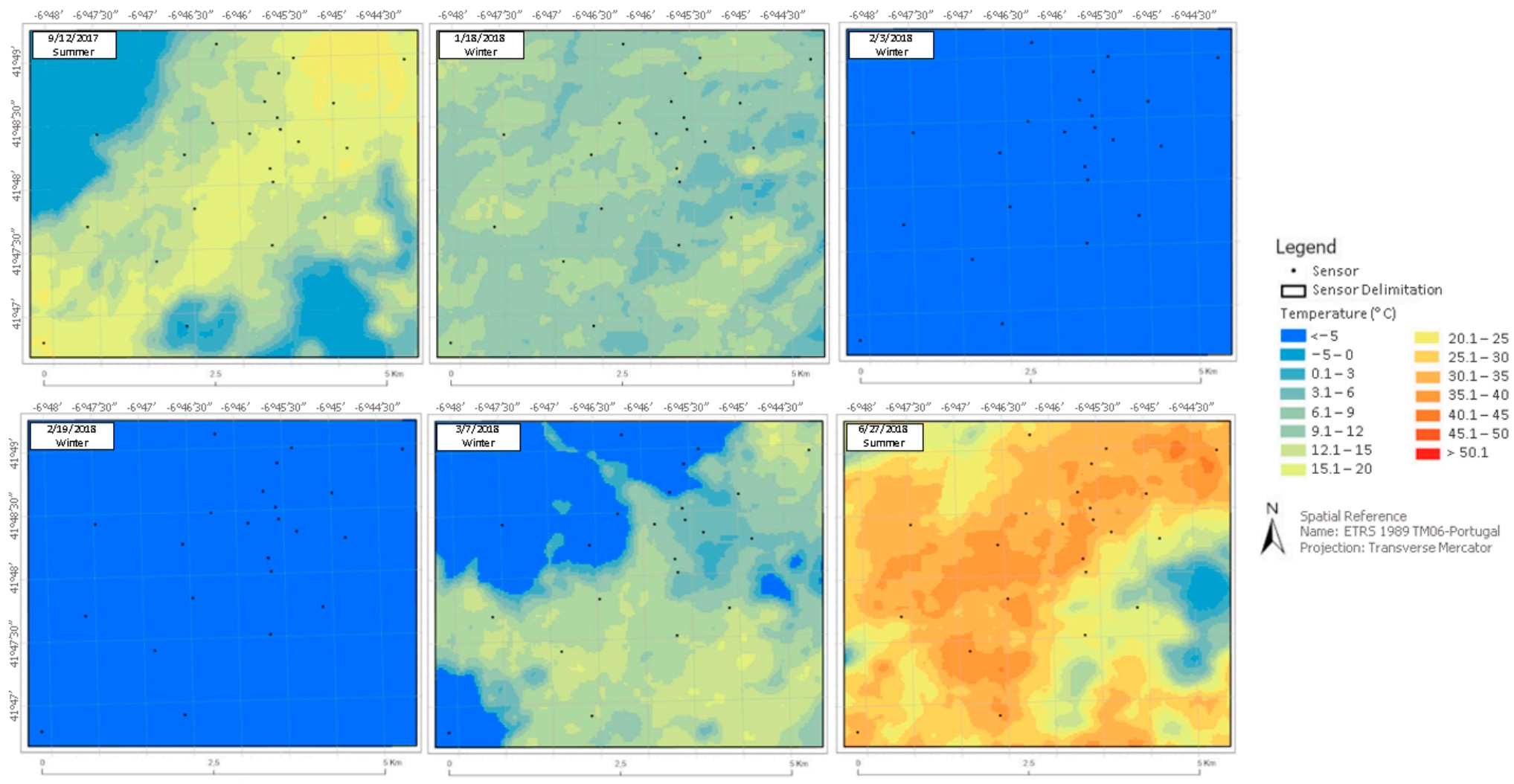

**Figure A1.** *Cont.*

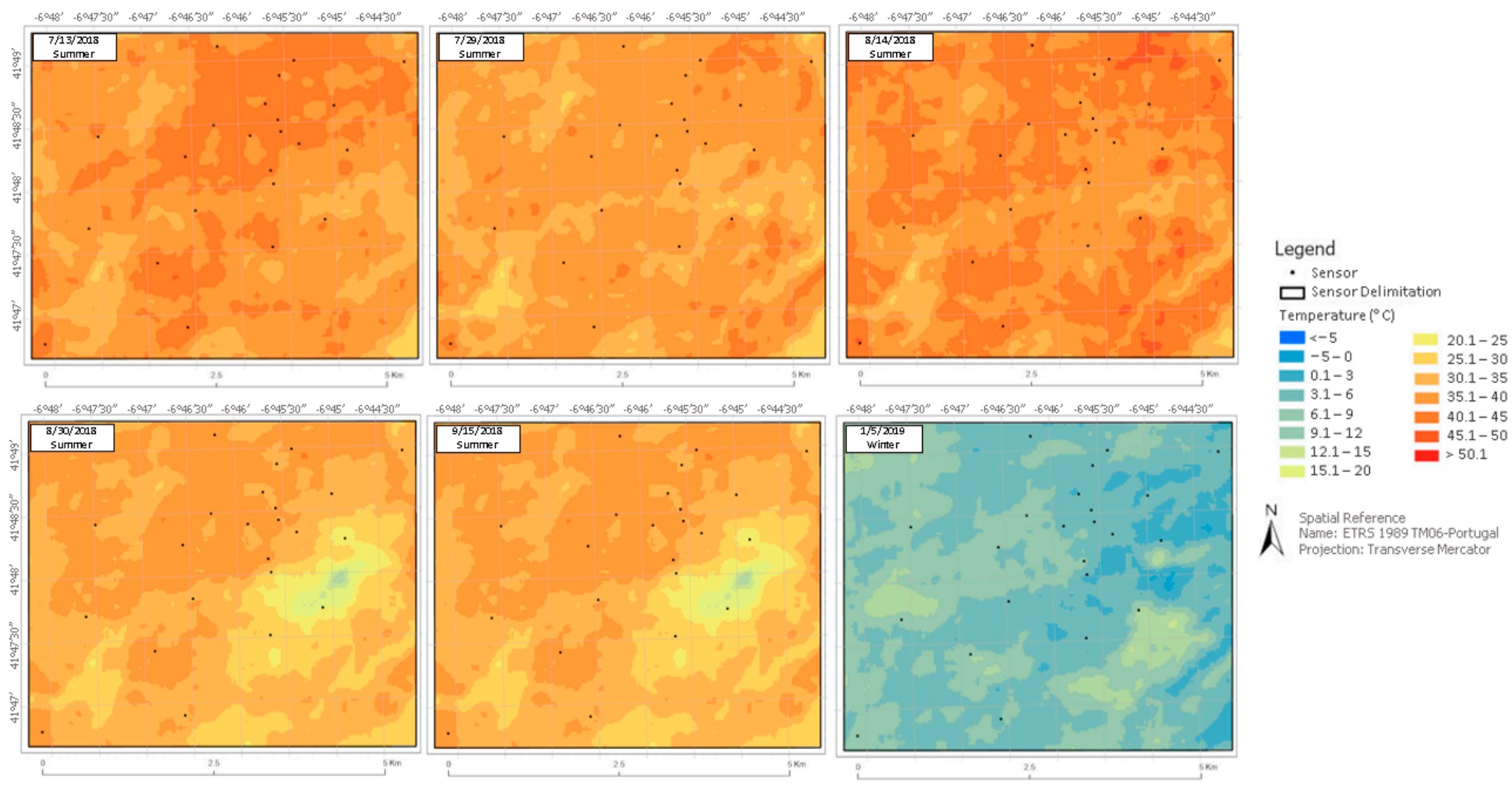

**Figure A1.** *Cont.*

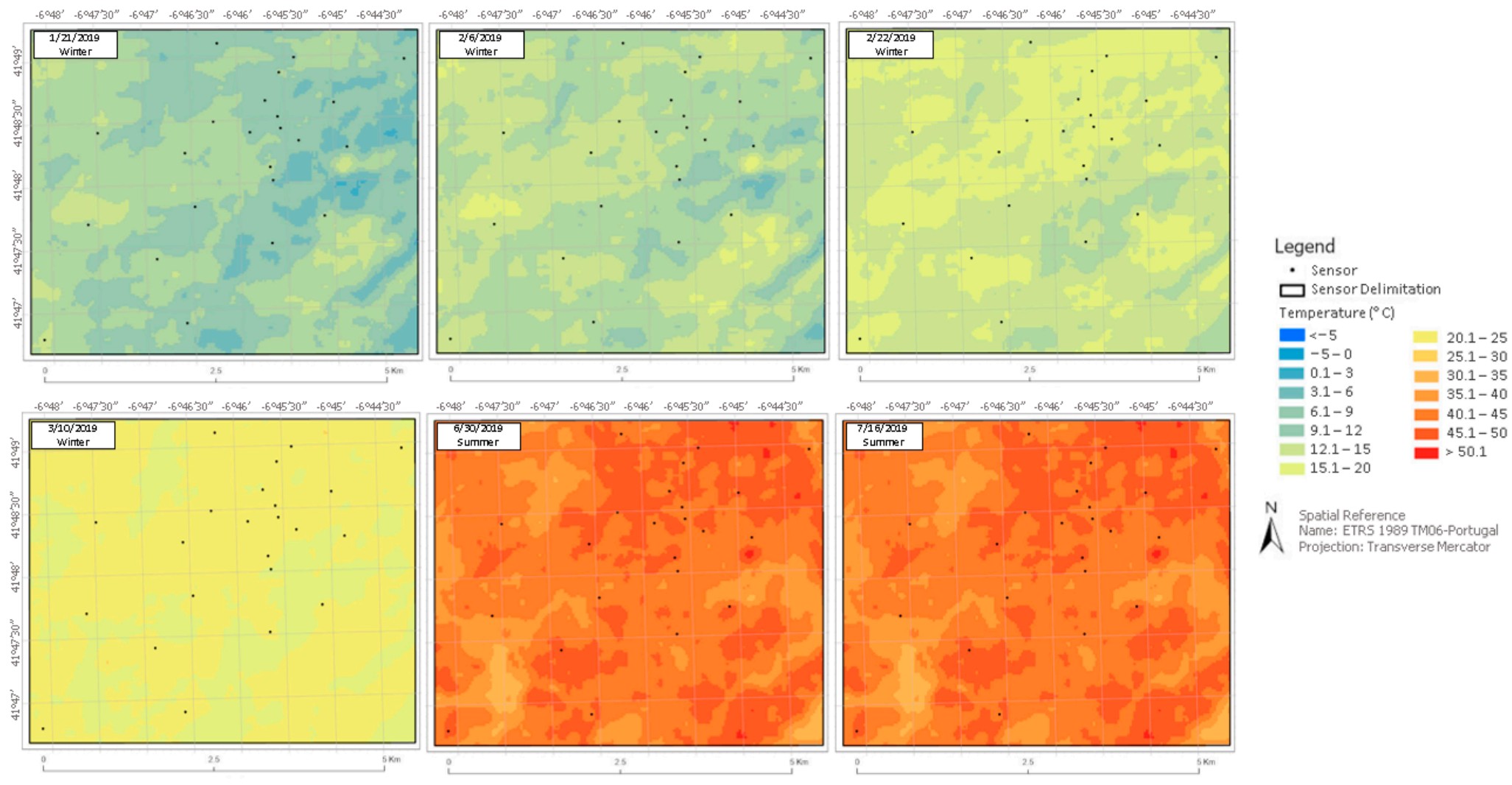

**Figure A1.** *Cont.*

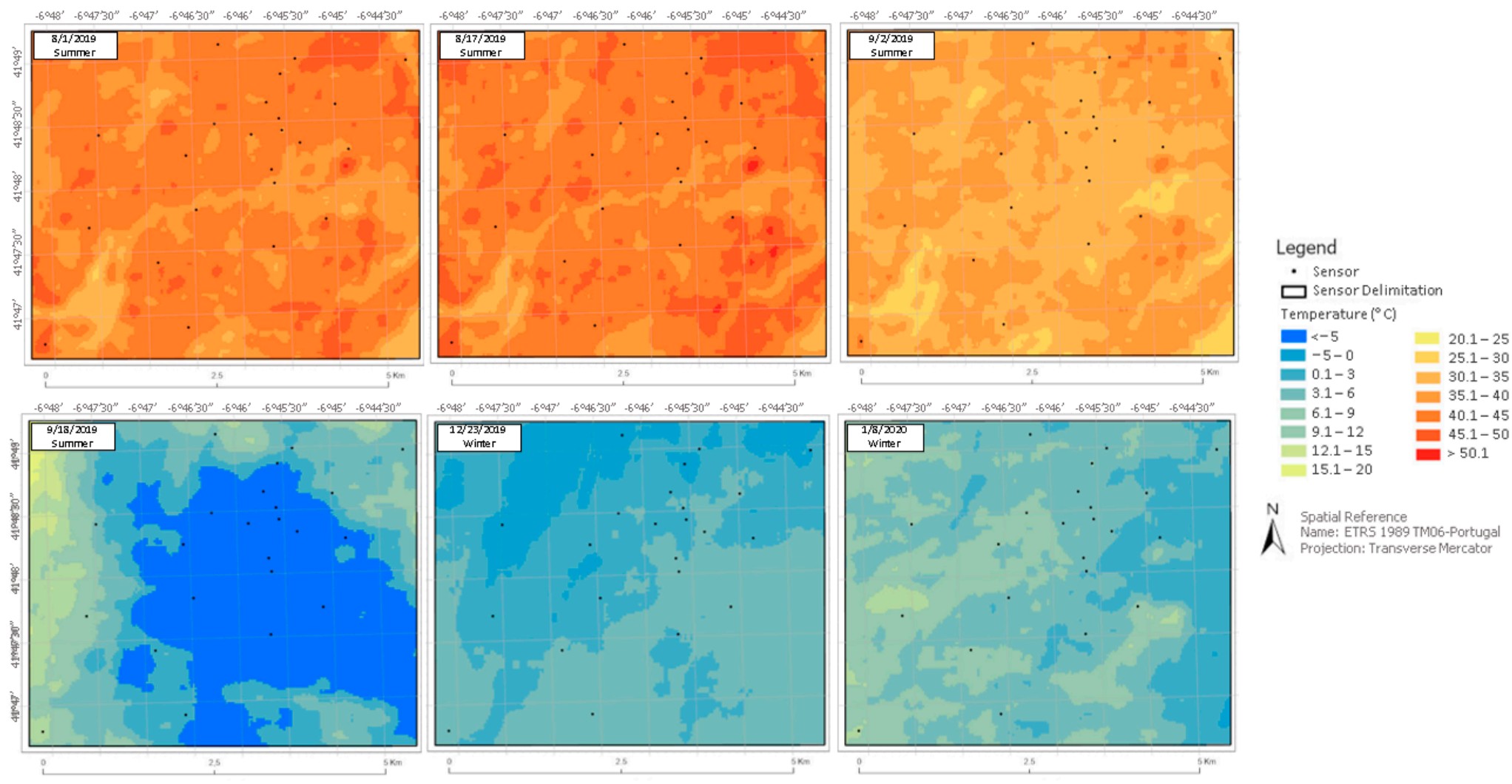

**Figure A1.** *Cont.*

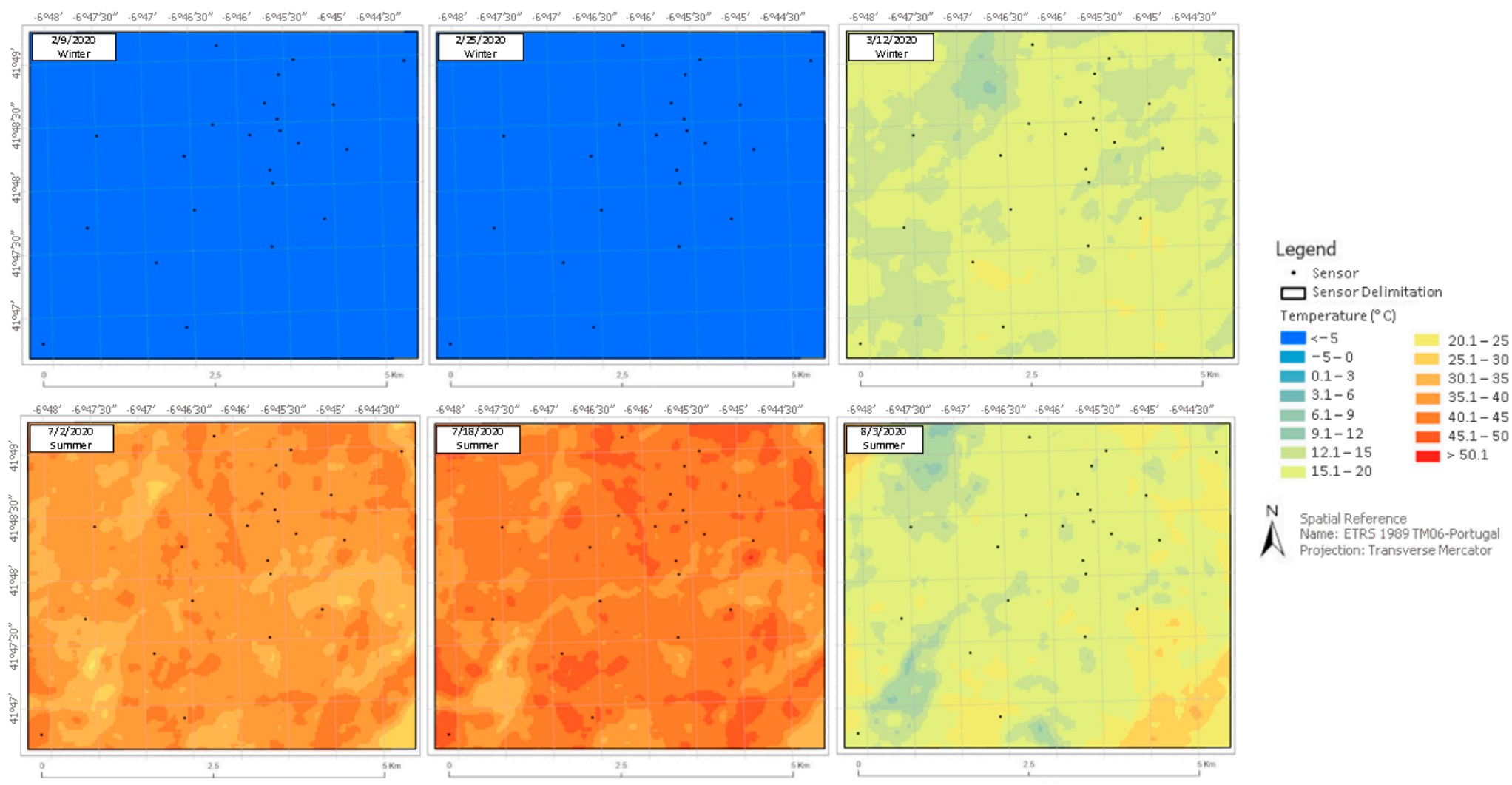

**Figure A1.** *Cont.*

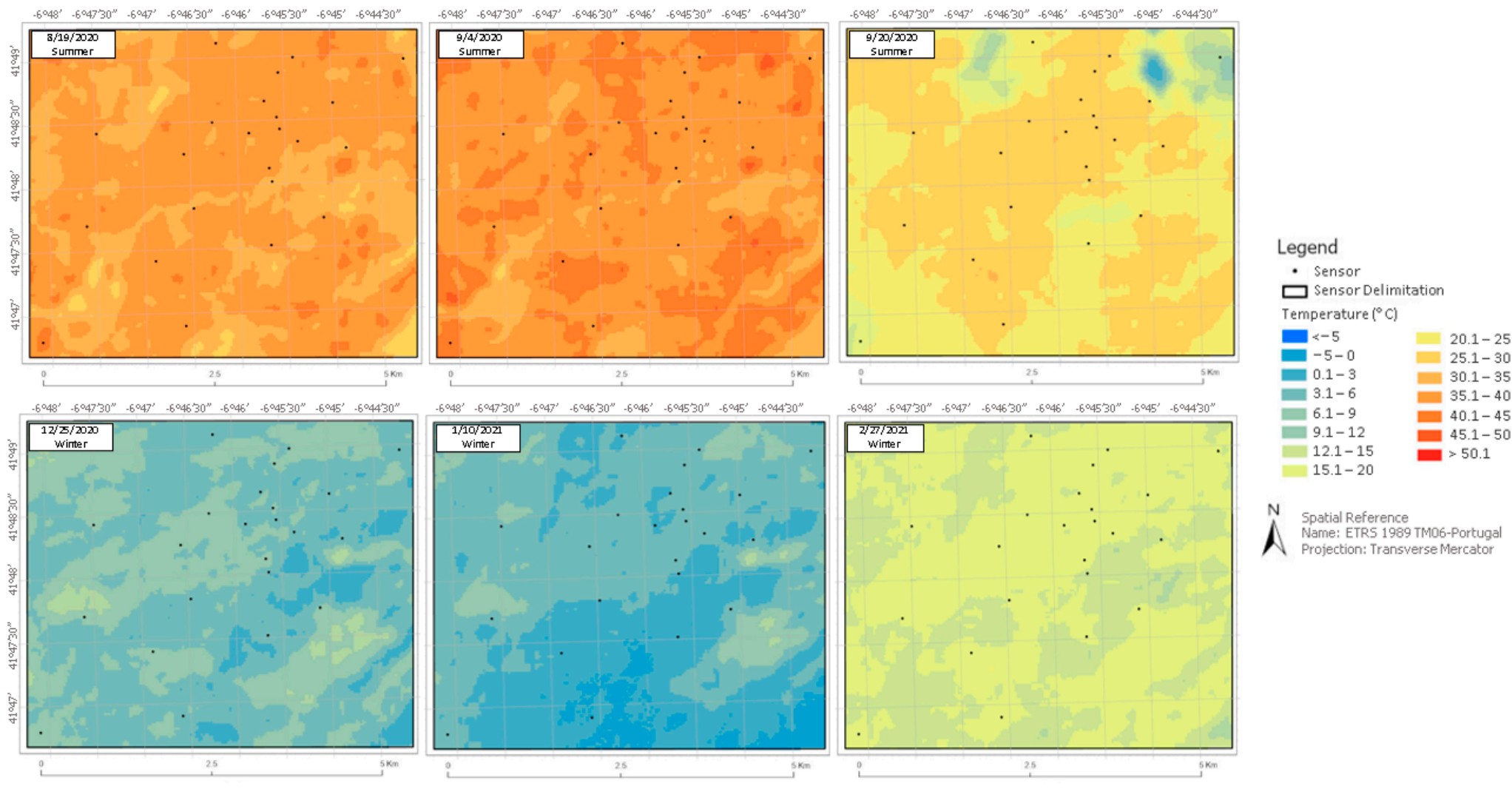

**Figure A1.** *Cont.*

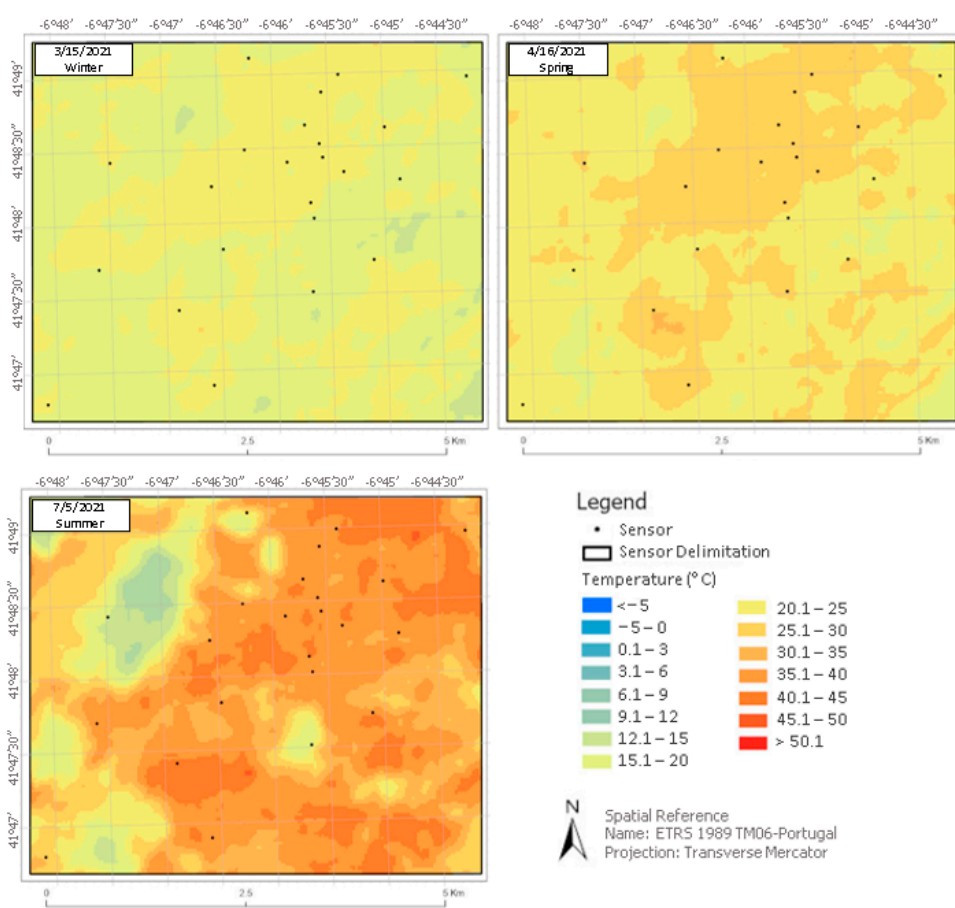

**Figure A1.** LST$_{L8}$ maps are generated in GEE.

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
