# Peer review of "Remote Sensing Image-Based Analysis of the Urban Heat Island Effect in Bragança, Portugal"

_environments, doi:10.3390/environments9080098_

Round 1

Reviewer 1 Report

Various studies have been conducted to analyze the urban heat island phenomenon. However, the overall flow of study is not systematic, and the readability is very poor. And what differentiates it from previous studies? There are many remote sensing-based heat island studies. However, it is not clear what the difference between this study and previous studies is. This comparison and review with previous studies also seems very important.

Author Response

Please, see the file attached.

Reviewer 2 Report

The paper ‘Remote Sensing Image-Based Analysis of the Urban Heat Is-2 land Effect in Bragança, Portugal’ concerns the evaluation of the urban heat island phenomenon in the mountainous city of Bragança, considering various climate zones in the discussed area. The authors described in details the evaluation of the phenomenon, using various statistical methods. The paper concerns an interesting problem and is worth publishing in the journal, after implementing several remarks.

Major issues:

-          The most important gap of the paper is missing the discussion section. The paper should contain the reference to other studies and compare the obtained results with other outcomes related to the urban heat island research. This part should be definitely implemented to the paper.

-           

-          The conclusions should be expresses more clearly and present the exact outcomes of the paper. In the current version, the information in the 'Conclusions' section refers more to the summary of the paper than to the presentation of the conclusions.

Minor issues:

-          In the Introduction section, pointing out the potential, practical use of the results would be beneficial for readers. The phenomenon of the urban heat island is currently one of the most important natural factors affecting inhabitants in municipalities. Therefore, the indication of such issues could increase the value of the paper.

-          Line 273: The authors conclude that the summer is characterized by the highest values of air temperature. Thus, I suggest changing the fragment of  ‘…compared to winter’ to ‘…significantly higher than in winter’.

-          As the analysis concern a single municipality, it would be beneficial to add some sentences concerning potential measures focused on the mitigation of the urban heat island occurrence in Bragança, especially in the densely developed areas with a high percentage of the artificial surface.

Author Response

Please, see the file attached.

Reviewer 3 Report

This paper aimed to analyze the behavior of UHI in different Local Climate Zones (LCZ) in Bragança city (Portugal), using Air Temperature (Ta), satellite images (Landsat 8), and on-site data. Besides, the objectives of this paper are: 119 compare SUHI and UHI data; analyze data collection on-site, to evaluate their possible 120 correlations; and apply descriptive and quantitative statistical analyses, to analyze the behavior of Bragança's UHI in each Local Climate 122 Zones (LCZ), in summer and winter. Therefore, in terms of research themes and purposes, research results should have academic and practical reference value. Nevertheless, The authors are advised to consider the following comments:

1.     Please provide additional information on why this study area was chosen? Is "Bragança" particularly suitable as a field for the demonstration of this research method?

2.     The paper mentions "It was not possible to perform any collection in the winter due to the weather conditions (precipitation and/or clouds on data) and due to the pandemic period (COVID-19)," in lines 162-165, if So, how to carry out the "data processing" of the subsequent winter images?

3.     Please explain how many of the 87 satellite images are in summer and in winter?

4.     In this study, the 27 surfaces were classified into six classes, namely: i) Asphalt (As); ii) Sidewalk (S); iii) Vegetation (V); iv) Vegetation with exposed Soil (Vs); v) Dry Vegetation (DV), and vi) Dry Vegetation and Rock (DVSt) (Table 2). Please provide additional information on why man-made environmental areas such as urban areas, buildings and roads are not included in the classification?

5.     The author mentioned that Equation 1 and Equation 2 (mistyped as Equation 22) in this study have been confirmed by the literature (32 and 53), but after reviewing these two literatures, one of them is a full report of this paper, and the other A literature does not directly confirm these two equations, the author is asked to confirm.

6.     The authors mention in line 262 of the paper "Wind speed was also analyzed in four different LCZs (LLR, GAB, CLR, and CM), to assess whether the urban context influenced the local circulation. At these same points, the analysis of Tr .", please explain how to deal with the two categories of OM and SB?

7.     The author is requested to add "Legends" to Figures 3 and 4, and specifically explain the meanings of the circles in the figures.

8.     Please mention in "Correlation Between TaSN and LSTL8": "In winter, all correlations were "Very Strong", which may be associated with lower solar incidence and shadow projections compared to summer", please explain how to calculate without winter Ta data and infer this result?

9.     The content written by the author in the conclusion is mostly common sense in this research field, and does not explain the important results of this research (only the research methods and steps are stated), and the author is advised to readjust it.

Author Response

Please, see the file attached.

Round 2

Reviewer 1 Report

I think that it is possible to publish the thesis because I think that it has been revised in many parts.

Author Response

Dear reviewer, good morning.

Thank you for the positive comments on the paper. Certainly, the recommendations you requested contributed to the final result.
We made only one additional change in the first paragraph of the abstract at the editor's request, which we hope you will find aligned with the rest of the paper.

Any questions, we are at your disposal.

Best regards,

Cátia Rodrigues de Almeida

Reviewer 2 Report

The authors considered all of my remarks. Therefore, the paper is now eligible to be published in the journal.

Author Response

(The authors gave the same response as above.)

Reviewer 3 Report

The author has responded and revised my review comments.  Current manuscript proposals may be accepted for publication.

Author Response

(The authors gave the same response as above.)
